# Medial anterior prefrontal cortex stimulation downregulates implicit reactions to threats and prevents the return of fear

**Eugenio Manassero¹†, Giulia Concina¹†, Maria Clarissa Chantal Caraig¹, Pietro Sarasso², Adriana Salatino², Raffaella Ricci², Benedetto Sacchetti¹\***

¹Rita Levi-Montalcini Department of Neurosciences, University of Turin, Turin, Italy; ²Department of Psychology, University of Turin, Turin, Italy

**Abstract** Downregulating emotional overreactions toward threats is fundamental for developing treatments for anxiety and post-traumatic disorders. The prefrontal cortex (PFC) is critical for top-down modulatory processes, and despite previous studies adopting repetitive transcranial magnetic stimulation (rTMS) over this region provided encouraging results in enhancing extinction, no studies have hitherto explored the effects of stimulating the medial anterior PFC (aPFC, encompassing the Brodmann area 10) on threat memory and generalization. Here we showed that rTMS over the aPFC applied before threat memory retrieval immediately decreases implicit reactions to learned and novel stimuli in humans. These effects enduringly persisted 1 week later in the absence of rTMS. No effects were detected on explicit recognition. Critically, rTMS over the aPFC resulted in a more pronounced reduction of defensive responses compared to rTMS targeting the dorsolateral PFC. These findings reveal a previously unexplored prefrontal region, the modulation of which can efficiently and durably inhibit implicit reactions to learned threats. This represents a significant advancement toward the long-term deactivation of exaggerated responses to threats.

**\*For correspondence:**
benedetto.sacchetti@unito.it

†These authors contributed equally to this work

**Competing interest:** The authors declare that no competing interests exist.

## Editor's evaluation

This study presents a valuable finding that rTMS over the aPFC, applied before threat memory retrieval, can immediately decrease implicit reactions to learned and novel stimuli in humans and that these effects can persist one week later, in the absence of rTMS. The evidence supporting the claims of the authors is solid and raise important hypotheses for further research. The work will be of interest to researchers in the fields of neuromodulation and affective neuroscience.

## Introduction

Emotional memories related to past threat experiences allow humans to predict future dangers and trigger adaptive defensive reactions when encountering learned threat-signaling cues (*DiFazio et al., 2022*). However, extremely dangerous situations may lead to psychological disorders (*Wilker et al., 2014*). Furthermore, the ability to generalize defensive reactions to new stimuli enables organisms to anticipate potential threats and respond to them based on similar perilous experiences lived in the past. On the other hand, evaluation mechanisms excessively biased toward threat generalization (i.e. overgeneralization) may underlie anxiety disorders and trauma (*Dunsmoor and Paz, 2015*). At the base of these processes, in a previous work (*Manassero et al., 2019*). we observed that autonomic-implicit and cognitive-explicit tunings may diverge when humans are exposed to the same new stimuli,

where cognitive generalization may enable a flexible evaluation of incoming cues to develop adaptive predictions of potential dangers. The crosswise presence of overgeneralization in anxiety diseases and the dissociation between autonomic and cognitive defensive response patterns highlight the importance of including both implicit and explicit generalization tasks to characterize fear-related processes in humans.

Attempting to downregulate the emotional overreactions toward threat-predictive and new stimuli is one of the main routes for developing effective treatments for anxiety and post-traumatic disorders. Common approaches such as pharmacological treatments and cognitive-behavioral therapy have demonstrated partial efficacy (*Taylor et al., 2012*), and recent evidence suggests that the functional outcome of behavioral methods may depend on the extent to which the prefrontal cortex is recruited during these processes (*Fonzo et al., 2017a*). Hence, new intervention strategies influencing the prefrontal dynamics would represent an important advance in the field (*Marković et al., 2021*).

Previous studies adopted transcranial direct current stimulation (tDCS) or transcranial electrical stimulation (tES) to disrupt the consolidation of these memories (*Asthana et al., 2013*; *Mungee et al., 2014*; *Mungee et al., 2016*), potentiate extinction processes (*Abend et al., 2016*; *van 't Wout et al., 2016*), and narrow threat generalization patterns (*Roesmann et al., 2022*), leading to contradictory results. According to one work (*Asthana et al., 2013*), cathodal stimulation over the dorsolateral prefrontal cortex (dlPFC) disrupted threat memory consolidation, with no enhancing effect of anodal stimulation. In contrast, other studies found an increase in implicit responses with anodal stimulation (*Mungee et al., 2014*) and no effect of cathodal stimulation (*Mungee et al., 2016*) over the same site. Moreover, one study employing anodal stimulation over the dlPFC (*van 't Wout et al., 2016*) revealed an improvement in extinction learning but no delayed effects on the recall of the extinction memory. A further investigation (*Abend et al., 2016*) reported that low-frequency alternating-current (AC) stimulation of the medial prefrontal cortex (mPFC) augmented the defensive responses, whereas direct-current (DC) stimulation widened threat generalization profiles.

An alternative neurostimulation approach is repetitive transcranial magnetic stimulation (rTMS), which ensures greater focality (*Miniussi et al., 2008*; *Elder and Taylor, 2014*). Some rTMS studies targeted the mPFC (*Guhn et al., 2014*) and the posterior PFC (*Raij et al., 2018*) to obtain a successful enhancement of extinction learning, while others (*Borgomaneri et al., 2020*; *Su et al., 2022*) targeted the dlPFC to disrupt threat-memory reconsolidation. Indeed, most rTMS-based research targeting the PFC has pursued an improvement of fear extinction, which may be followed by a return of fear with a change of context (i.e., renewal) (*Vervliet et al., 2013*) where prevention of relapse over time is the main challenge for therapies dedicated to post-traumatic and anxiety disorders. No previous studies reported significant effects in downmodulating the defensive responses triggered by a learned threatening stimulus without adopting fear extinction.

So far, human brain stimulation studies have been mainly focused on the dorsolateral region of the PFC (*Marković et al., 2021*), partly because other prefrontal areas involved in the top-down regulation of subcortical threat-detection systems – such as the ventromedial PFC (vmPFC), are too deep to be reached with TMS (*Raij et al., 2018*). However, within the PFC, a brain structure that is emerging to be engaged in downstream emotional regulation is the anterior prefrontal cortex (aPFC), also known as the frontopolar cortex or rostral frontal cortex. The aPFC encompasses the most anterior portion of the prefrontal cortex (Brodmann area 10 [BA 10]) (*Ramnani and Owen, 2004*) and extends over a wider cortical space in humans than in other species (*Semendeferi et al., 2001*). Even if it has not been included in fear network models so far, many studies (*Volman et al., 2013*; *Koch et al., 2018*; *Bramson et al., 2020*) highlighted its role in emotional downregulation. Anatomical projections have been found between the lateral (*Bramson et al., 2020*; *Folloni et al., 2019*) and the medial aPFC (*Peng et al., 2018*) and the amygdala, and functional connectivity has been detected between the aPFC and the vmPFC during fear downregulation (*Klumpers et al., 2010*). Notably, hypoactivation, reduced connectivity, and altered thickness of aPFC were reported in post-traumatic stress disorder (PTSD) patients (*Lanius et al., 2005*; *Morey et al., 2008*; *Sadeh et al., 2015*; *Sadeh et al., 2016*), whereas a longitudinal study (*Kaldewaij et al., 2021*) showed that strong activation of the aPFC resulted in a higher resilience against PTSD onset. Accordingly, enhanced aPFC activity and potentiated aPFC-vmPFC connectivity were detected after an effective therapy in PTSD patients (*Fonzo et al., 2017b*). Crucially, the aPFC is a surface area easily accessible with rTMS. However, to our knowledge, no study has been conducted so far to explore

the effects of aPFC stimulation on the expression of a threat memory without extinction learning in humans.

In the current study, we posited that applying rTMS to the aPFC could influence implicit defensive responses to a learned threat-predictive stimulus and/or the conscious recognition of it. Subsequently, we explored additional hypotheses. The second hypothesis centered on the potential extension of rTMS dampening effects to new stimuli, thereby reducing threat generalization. The third hypothesis focused on the enduring persistence of rTMS effects on defensive responses over time. The final hypothesis proposed that the dampening effects achieved by stimulating the aPFC might surpass those observed when targeting the dorsolateral PFC.

## Results

### aPFC-focused rTMS effects on implicit defensive reactions toward threat-predictive and new cues

To explore the effects of an aPFC-centered rTMS on the implicit responses to a learned threat, we designed a three-session experiment starting with a threat learning session followed by an implicit retention test and a follow-up implicit retest (*Figure 1*).

During the learning session, participants learned to associate an auditory cue (conditioned stimulus [CS], 800 Hz) with a mild electric stimulation (unconditioned stimulus [US], individually calibrated intensity) in a given environment (context A). We adopted a single-cue learning paradigm because it more ecologically reflects real-life traumatic experiences (*Resnik and Paz, 2015*; *Wong and Lovibond, 2017*; *Grosso et al., 2018*; *Concina et al., 2018*; *Grosso et al., 2017*). To validate the between-groups homogeneity in the painful stimuli perception, we compared the post-conditioning US ratings and observed no significant differences between groups (Student's unpaired $t$-test, $t_{(58)}$ = 0.799, p=0.428, $\eta_p^2$ = 0.011) (*Table 1*). We also did not observe significant differences between groups in skin conductance responses (SCRs) to the CS during the preconditioning phase ($t_{(58)}$ = 0.418, p=0.677, $\eta_p^2$ = 0.003), to the CS during the conditioning phase (2 × 15 mixed ANOVA; main effect of group: $F_{(1,52)}$ = 2.367, p=0.130, $\eta_p^2$ = 0.044; main effect of trial: $F_{(8.762,455.600)}$ = 13.366, p<0.001, $\eta_p^2$ = 0.204; group × trial interaction: $F_{(8.762,455.600)}$ = 1.619, p=0.109, $\eta_p^2$ = 0.030; Student's unpaired $t$-test on the averaged response, $t_{(58)}$ = 1.290, p=0.202, $\eta_p^2$ = 0.028), nor to the US during the conditioning phase ($t_{(58)}$ = 1.011, p=0.316, $\eta_p^2$ = 0.017) (*Figure 2—figure supplement 1*).

One week later, we tested the implicit memory of the learned association in control sham-stimulated subjects and in those who received rTMS over the aPFC shortly before the memory test. To locate this brain region, which corresponds to the BA 10 (*Hanlon et al., 2018*), we positioned the coil over the frontopolar midline electrode (Fpz) adopting the international 10–20 electroencephalogram (EEG) coordinate system (*Jasper, 1958*) since previous rTMS studies (*Guhn et al., 2014*; *Herrmann et al., 2017*; *Karmann et al., 2016*) ensured this placement reached the aPFC. An offline 10 min session of 1 Hz-rTMS targeting this neural site (aPFC, n = 30) was applied immediately before memory retrieval (*Figure 2A*). Control subjects underwent a 10 min sham stimulation procedure over the same cortical area (sham, n = 30).

Memory retention was tested in a different environment from that where the learning had occurred (context B) to avoid any contextual influence on retrieval (*Manassero et al., 2019*; *Ameli et al., 2001*; *Maren et al., 2013*; *Sacco and Sacchetti, 2010*; *Sacchetti et al., 1999*). Indeed, the context shift for this session mirrors a real-life treatment setting, which unlikely takes place in the threatening location. To test implicit threat memory, we performed an implicit recognition task in which subjects were exposed to the CS while being recorded in their evoked autonomic reactions (i.e., electrodermal SCRs). No US shocks were delivered during this phase. Besides the CS, participants were presented with two novel but perceptually similar tones (NS$_1$, 1000 Hz; NS$_2$, 600 Hz) to study threat generalization. Auditory frequencies of NSs were selected to obtain a slowly decaying gradient of defensive tunings (*Manassero et al., 2019*; *Onat and Büchel, 2015*; *Laufer et al., 2016*). To test the effects of rTMS on memory retention, we compared the between-group differences as well as the within-group differences from the acquisition phase to the testing phase through a 2 × 2 mixed ANOVA. This analysis yielded a nonsignificant main effect of group ($F_{(1,58)}$ = 2.015, p=0.161, $\eta_p^2$ = 0.034), a nonsignificant main effect of phase ($F_{(1,58)}$ = 0.053, p=0.818, $\eta_p^2$ = 0.001), and a significant group × phase interaction ($F_{(1,58)}$ = 13.445, p=0.001, $\eta_p^2$ = 0.188). Simple main effects analysis

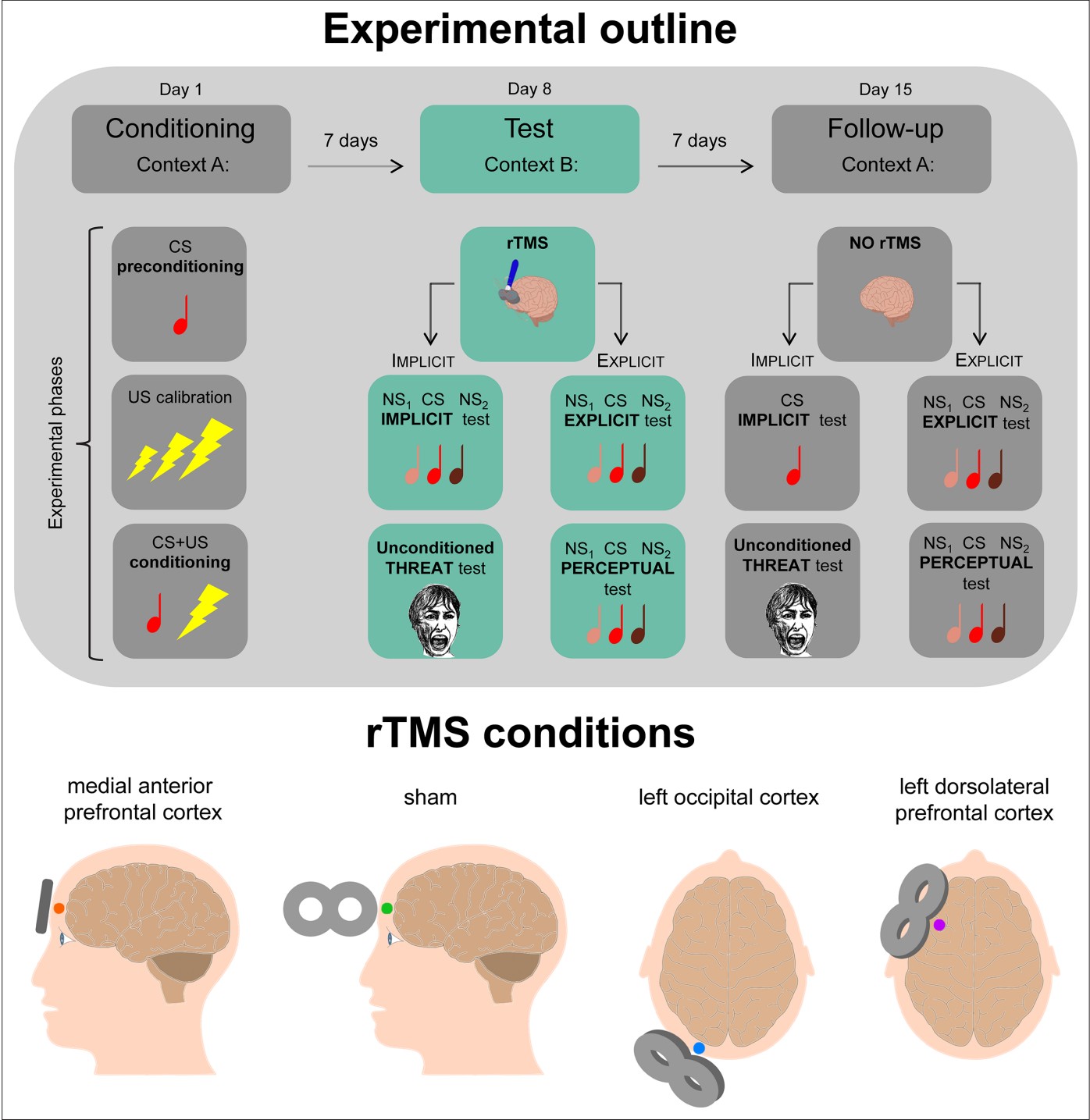

**Figure 1.** Schematic diagram depicting the experimental outline and repetitive transcranial magnetic stimulation (rTMS) conditions. In the first session (day 1, context A), participants underwent a single-cue threat conditioning in which a tone (CS) was paired with a mild electrical shock (US). In the second session (day 8, context B), a 1 Hz-rTMS procedure was actively applied over the medial anterior prefrontal cortex (aPFC, n = 30; aPFC-E, n = 21), sham-applied over the same site (sham, n = 30; sham-E, n = 21), actively applied over the left occipital cortex (OC, n = 30) and over the left dorsolateral prefrontal cortex (dlPFC, n = 30). In the implicit conditions (aPFC, sham, OC, dlPFC), subjects underwent an implicit test during which they were presented with the CS and two new stimuli (NS$_1$ and NS$_2$) and then an unconditioned threat test while being recorded in their skin conductance responses (SCRs). In the explicit conditions (aPFC-E, sham-E), participants underwent an explicit two-alternative forced-choice (2AFC) recognition task during which they were presented with tone pairs each composed of the CS and one of the two NSs, and they were asked to recognize the CS providing a confidence level for each choice. Last, participants underwent a 2AFC perceptual discrimination test, in which they had to judge whether the two tones in each pair (CS and/or NSs) were 'the same tone' or 'different tones'. The third session (day 15, context A) was identical to the second one except for the absence of the rTMS.

**Table 1.** Experimental groups' descriptive, experimental, and clinical data.

The table reports, for each experimental condition, sample size (*N*), sex distribution (F = female, M = male), mean age, State-Trait Anxiety Inventory Form Y (STAI-Y) State subscale score during session 1 (S1), session 2 (S2), and session 3 (S3), and Trait subscale score, US current intensity (mA), post-conditioning US rating, rTMS resting motor threshold (rMT), rTMS power, and discomfort stimulation (DS) current intensity (mA). All data are mean ± standard deviation.

| Group | N | Sex | Age | STAI-Y State (S1) | STAI-Y State (S2) | STAI-Y State (S3) | STAI-Y Trait | US (mA) | US rating | rTMS rMT | rTMS power | DS (mA) |
|---|---|---|---|---|---|---|---|---|---|---|---|---|
| aPFC | 30 | 18 F 12 M | 24.45 ± 3.78 | 30.97 ± 4.07 | 32.47 ± 7.16 | 30.60 ± 6.04 | 39.27 ± 6.18 | 4.92 ± 2.06 | 5.28 ± 0.90 | 58.20 ± 6.40 | 39.73 ± 1.11 | - |
| Sham | 30 | 18 F 12 M | 23.35 ± 2.35 | 33.23 ± 5.86 | 32.70 ± 7.74 | 31.87 ± 6.51 | 38.77 ± 4.02 | 4.88 ± 2.45 | 5.47 ± 0.88 | - | - | - |
| OC | 30 | 18 F 12 M | 24.14 ± 2.62 | 32.33 ± 5.51 | 31.53 ± 7.57 | 30.60 ± 6.75 | 39.03 ± 5.12 | 4.99 ± 3.17 | 5.28 ± 1.06 | 60.90 ± 6.67 | 39.70 ± 1.47 | - |
| dlPFC | 30 | 18 F 12 M | 23.91 ± 3.15 | 31.70 ± 5.40 | 30.83 ± 7.04 | 30.13 ± 5.88 | 39.17 ± 5.85 | 5.16 ± 2.43 | 5.57 ± 1.45 | 58.77 ± 5.89 | 39.90 ± 0.40 | - |
| aPFC-E | 21 | 13 F 8 M | 24.39 ± 2.43 | 31.71 ± 4.89 | 30.90 ± 5.66 | 30.48 ± 4.96 | 38.29 ± 6.21 | 5.13 ± 1.86 | 5.43 ± 0.94 | 58.67 ± 7.16 | 39.52 ± 1.54 | - |
| Sham-E | 21 | 13 F 8 M | 23.83 ± 2.73 | 33.10 ± 5.59 | 31.48 ± 5.54 | 30.38 ± 7.73 | 38.29 ± 5.22 | 5.27 ± 3.19 | 5.31 ± 1.31 | - | - | - |
| Ctrl discomfort | 10 | 5F 5M | 22.34 ± 3.67 | 34.40 ± 4.20 | 36.50 ± 6.47 | 34.20 ± 5.98 | 39.70 ± 4.03 | 6.97 ± 4.14 | 5.65 ± 1.11 | - | - | 6.65 ± 2.25 |

aPFC = anterior prefrontal cortex; dlPFC = dorsolateral prefrontal cortex; rTMS = repetitive transcranial magnetic stimulation; US = unconditioned stimulus; OC = occipital cortex.

revealed no significantly different mean CS-evoked SCRs between groups during the conditioning phase (p=0.506; Bonferroni corrected). On the contrary, during the test phase subjects who received rTMS over the aPFC exhibited weakened CS-related SCRs than those observed in the sham group (p=0.006; Bonferroni corrected). Moreover, the aPFC group showed reduced autonomic responses to the CS from conditioning to test (p=0.008; Bonferroni corrected), whereas the sham group displayed increased mean SCRs to the CS from conditioning to test (p=0.018; Bonferroni corrected) (*Figure 2B and C*). This data indicates that the rTMS procedure affected SCRs triggered by memory retrieval performed shortly after rTMS. To the best of our knowledge, this is the first evidence that brain stimulation may promptly attenuate implicit defensive reactions during memory retrieval.

In the test session, we also analyzed threat generalization to the NSs through a 2 × 3 mixed ANOVA, which showed a significant main effect of group ($F_{(1,58)}$ = 5.310, p=0.025, $\eta_p^2$ = 0.084), a nonsignificant main effect of tone ($F_{(2,116)}$ = 0.690, p=0.504, $\eta_p^2$ = 0.012), and a nonsignificant group × tone interaction ($F_{(2,116)}$ = 1.301, p=0.276, $\eta_p^2$ = 0.022), revealing that the aPFC group displayed overall attenuated responses to tones relative to the sham condition (*Figure 2D and E*).

We next sought to disambiguate whether the rTMS effects were due to a general downregulation of electrodermal responsivity or whether they specifically targeted the threat memory. To this end, subjects were presented with an unconditioned threatening stimulus consisting of a female scream sample (unconditioned stimulus 2 [$US_2$]) while being recorded in their SCRs. No significant differences emerged between conditions ($t_{(58)}$ = 0.334, p=0.739, $\eta_p^2$ = 0.002), indicating that the rTMS did not cause an overall inhibition of electrodermal reactivity (*Figure 2F*).

To test whether and to what extent rTMS-related outcomes endured beyond the aftereffect window and persisted over a long-term period, we planned a follow-up session. One week after the threat memory retrieval test, all participants returned to the conditioning room (context A) and underwent a re-testing phase, identical to the testing one except for the absence of rTMS administration. This phase also allowed us to test a possible renewal effect (*Vervliet et al., 2013*) since subjects were re-exposed to the original threatening environment.

Concerning the implicit responses to the CS, a 2 × 2 mixed ANOVA showed a nonsignificant main effect of group ($F_{(1,58)}$ = 1.952, p=0.168, $\eta_p^2$ = 0.033), a significant main effect of phase ($F_{(1,58)}$ = 7.690, p=0.007, $\eta_p^2$ = 0.117), and a significant group × phase interaction ($F_{(1,58)}$ = 9.966, p=0.003, $\eta_p^2$ = 0.147). Simple main effects analysis revealed that participants of the aPFC group persisted in displaying weaker SCRs than those observed in the sham group (p=0.006; Bonferroni corrected).

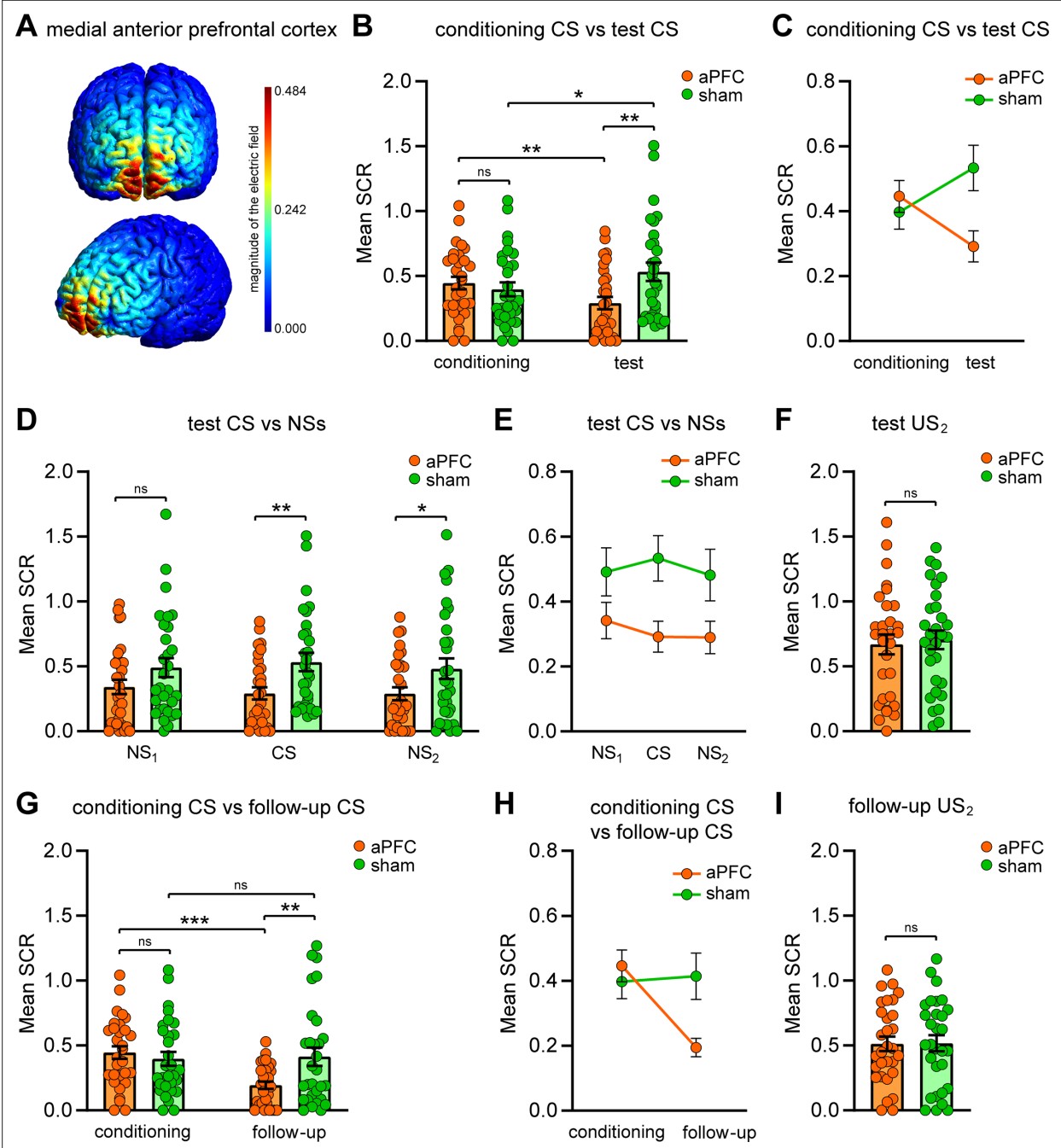

**Figure 2.** Effects of repetitive transcranial magnetic stimulation (rTMS) over the anterior prefrontal cortex (aPFC) on immediate and remote implicit threat memory, threat generalization to new stimuli, and overall electrodermal responsivity. (**A**) Simulation of rTMS effects on the neural tissue of the medial aPFC (medial Brodmann area 10 [BA 10]), performed with SimNIBS 4.0 software. The magnitude of the electric field is expressed in V/m. (**B, C**) Dot plot and line chart representing the mean skin conductance responses (SCRs) elicited by the CS during the conditioning session and test session in the two different conditions. Groups' reactions were not different during the conditioning phase, whereas during the test phase the group stimulated over the aPFC (n = 30) showed attenuated implicit reactions relative to the sham condition (n = 30). The aPFC group displayed reduced autonomic reactions to the CS from conditioning to test, while the sham group showed an increase in defensive responses. (**D, E**) Implicit reactions to all the tones (NS$_1$, CS, and NS$_2$) during the test session were decreased in the aPFC group relative to the sham group. Although we found a significant main effect of group and no group × tone interaction effect, we reported the statistical significance marks of simple main effects. (**F**) Implicit reactions to the US$_2$ during the test session were not different between conditions, showing no rTMS effects on the overall electrodermal responsivity. (**G, H**) In the follow-up session, the aPFC group enduringly demonstrated reduced implicit reactions to the CS relative to the sham group and to the conditioning phase. (**I**) Implicit reactions to the US$_2$ during the follow-up session were not different between groups. *p<0.05, **p<0.01, ***p<0.001. All data are mean and

*Figure 2 continued on next page*

*Figure 2 continued*

SEM. 2 × 2 mixed ANOVA followed by Bonferroni-adjusted post hoc comparisons (**B, C, G, H**); 2 × 3 mixed ANOVA followed by Bonferroni-adjusted post hoc comparisons (**D, E**); Student's unpaired *t*-test (**F, I**).

The online version of this article includes the following source data and figure supplement(s) for figure 2:

**Source data 1.** SCR raw data of the sham group during the conditioning, the test, and the follow-up.

**Figure supplement 1.** Implicit reactions during preconditioning (CS) and conditioning (CS, US) in the anterior prefrontal cortex (aPFC) and sham groups.

**Figure supplement 2.** Effects of a discomfort-inducing procedure on immediate and remote implicit threat memory.

**Figure supplement 2—source data 1.** SCR raw data of the ctrl discomfort group during the conditioning, the test, and the follow-up.

Moreover, the aPFC group persisted in showing a decrease of defensive reactions to the CS from conditioning to follow-up (p<0.001; Bonferroni corrected), while the sham group did not display significantly different SCRs in the two phases (p=0.787; Bonferroni corrected) (*Figure 2G and H*).

These findings support an enduring effect of the aPFC-rTMS in attenuating the long-term implicit defensive responses to the learned threat-predictive cue, even with the re-exposure to the environment where threat learning had occurred. We next analyzed the autonomous response patterns to the female scream sample (unconditioned stimulus 2 [$US_2$]) and again we found that reactions did not differ between groups ($t_{(58)}$ = 0.057, p=0.955, $\eta_p^2$ < 0.001) (*Figure 2I*). Thus, the persistent effect was expressed notwithstanding an unaffected electrodermal overall reactivity.

An important aspect to consider is that rTMS application over the forehead can be subjectively perceived as unpleasant. We, therefore, investigated whether an rTMS-related discomfort before memory retrieval might have provoked habituation to unpleasant stimulations, leading to a reduction in SCR levels during CS presentations. We repeated the entire experiment in one further group (ctrl discomfort, n = 10) by replacing the rTMS procedure with a 10 min discomfort-inducing procedure over the same site of the forehead to mimic the rTMS-evoked unpleasant sensations in the absence of neural stimulation effects. This group showed no significantly different CS-evoked SCR levels to those of the sham group during the test session as well as during the follow-up session (*Figure 2—figure supplement 2*). Thus, the discomfort experienced during the rTMS procedure did not contribute to the reduction of electrodermal responses observed in the aPFC-stimulated group.

## aPFC-focused rTMS effects on the explicit memory recognition and perceptual discrimination

We then investigated the effect of rTMS over the aPFC on the retention of explicit-declarative threat memories. A further group of subjects that received the identical 1 Hz-rTMS procedure over the aPFC (aPFC-E, n = 21) and a further control group (sham-E, n = 21) underwent an explicit two-alternative forced-choice (2AFC) recognition task, in which they were presented with a random sequence of tone pairs, each composed of the CS and one of the two NSs. Subjects were asked to consciously identify which stimulus of each pair was the one previously paired with the US (i.e., the CS) and to provide a subjective confidence level for each choice using a scale ranging from 0 (completely unsure) to 10 (completely sure) (*Manassero et al., 2019*; *Manassero et al., 2022*). Both groups reported nonsignificantly different post-conditioning US ratings ($t_{(40)}$ = 0.339, p=0.737, $\eta_p^2$ = 0.003) and successfully identified the CS amongst the NSs with an accuracy level above the 50% chance level (aPFC-E: $t_{(20)}$ = 9.226, p<0.001, $\eta_p^2$ = 0.810; sham-E: $t_{(20)}$ = 14.240, p<0.001, $\eta_p^2$ = 0.910). A between-groups comparison ($t_{(40)}$ = 1.114, p=0.272, $\eta_p^2$ = 0.030) showed no differences in the explicit recognition accuracy (*Figure 3A*). The two groups were not differently confident when making their choices ($t_{(40)}$ = 0.842, p=0.405, $\eta_p^2$ = 0.017) (*Figure 3B*), thereby supporting the lack of rTMS-related effects.

Next, since a previous study (*Roesmann et al., 2022*) targeting the vmPFC modulated perceptual discrimination processes, we implemented a 2AFC perceptual task in which we investigated the ability of participants to sensory discriminate between the CS and the two NSs by collecting binary 'same or different' judgments as well as confidence ratings. The perceptual discrimination test yielded no significant between-groups differences in accuracy ($t_{(40)}$ = 1.362, p=0.181, $\eta_p^2$ = 0.044) as well as confidence levels ($t_{(40)}$ = 0.917, p=0.365, $\eta_p^2$ = 0.021). Indeed, both groups discriminated the CS from the NSs with high precision (aPFC-E: 0.980 ± 0.015 SEM; sham-E: 1.000 ± 0.000 SEM) and with

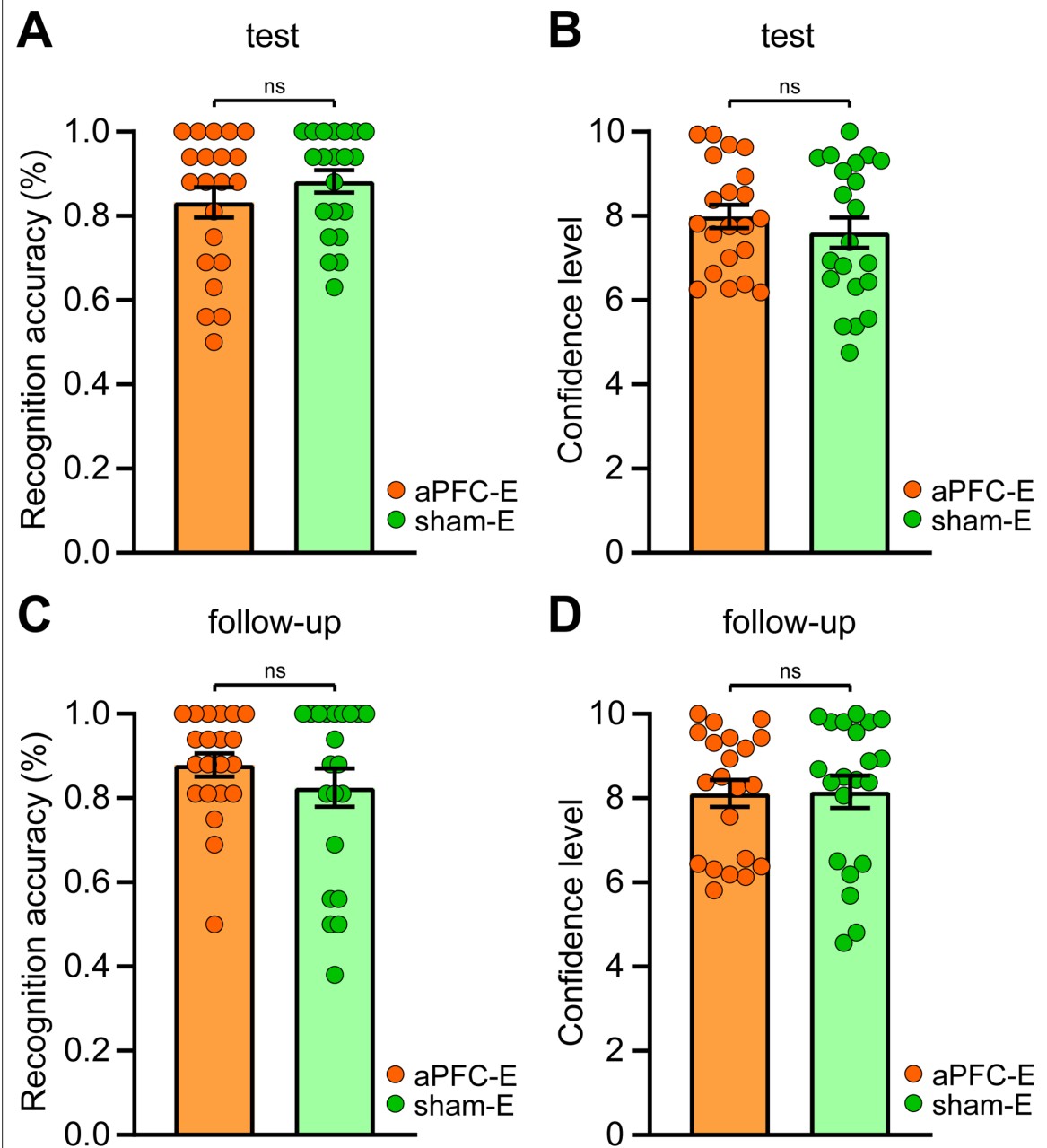

**Figure 3.** Effects of repetitive transcranial magnetic stimulation (rTMS) over the anterior prefrontal cortex (aPFC) on immediate and remote explicit threat memory. (**A**) During the test session, explicit recognition patterns were not different between the group stimulated over the aPFC (n = 21) and the sham group (n = 21). (**B**) During the test session, confidence ratings did not differ between the two conditions. (**C**) During the follow-up session, aPFC-E and sham-E groups identified the CS between the NSs in a not different manner. (**D**) During the follow-up session, aPFC-E and sham-E groups were not differently confident about their explicit choices. All data are mean and SEM. Student's unpaired *t*-test (**A–D**).

The online version of this article includes the following source data for figure 3:

**Source data 1.** Raw data of aPFC-E and sham-E groups during the explicit recognition tasks and the perceptual tasks.

no different confidence levels (aPFC-E: 9.409 ± 0.153 SEM; sham-E: 9.586 ± 0.117 SEM), thereby showing no rTMS effects on sensory abilities.

These data suggest that the pre-retrieval rTMS procedure over the aPFC did not affect the explicit recognition nor the perceptual discrimination of a learned threat.

During the follow-up session, explicit recognition patterns demonstrated an over-chance accuracy level for each group (aPFC-E: $t_{(20)} = 13.780$, p<0.001, $\eta_p^2 = 0.905$; sham-E: $t_{(20)} = 7.162$, p<0.001, $\eta_p^2$

= 0.720). Again, here there were no between-group differences ($t_{(40)}$ = 1.024, p=0.312, $\eta_p^2$ = 0.026) since both groups achieved a high recognition accuracy (**Figure 3C**). Groups did also not report different confidence levels ($t_{(40)}$ = 0.084, p=0.934, $\eta_p^2$ < 0.001) (**Figure 3D**).

As in the case of the previous session, we did not observe significant between-group differences in the perceptual discrimination ($t_{(40)}$ = 1.000, p=0.323, $\eta_p^2$ = 0.024) and the respective confidence ratings ($t_{(40)}$ = 0.149, p=0.882, $\eta_p^2$ < 0.001). Indeed, the discrimination accuracy (aPFC-E: 1.000 ± 0.000 SEM; sham-E: 0.993 ± 0.007 SEM) and the self-assessed confidence (aPFC-E: 9.598 ± 0.147 SEM; sham-E: 9.633 ± 0.182 SEM) were high in each condition.

## Topographical selectivity of rTMS effects on implicit defensive responses to threat-predictive and new cues

To ascertain the topographical selectivity, in one further condition (OC, n = 30) we applied the rTMS over the left occipital cortex as an active control site (**Figure 4A**) and contrasted its implicit reactions with those of the group stimulated over the aPFC.

No differences emerged between the two conditions in terms of post-conditioning US ratings ($t_{(58)}$ = 0.000, p=1.000, $\eta_p^2$ = 0.000) (**Table 1**), SCR responses to the CS during the preconditioning phase ($t_{(58)}$ = 1.037, p=0.304, $\eta_p^2$ = 0.018), to the CS during the conditioning phase (2 × 15 mixed ANOVA; main effect of group: $F_{(1,54)}$ = 0.124, p=0.726, $\eta_p^2$ = 0.002; main effect of trial: $F_{(9.368,505.856)}$ = 13.341, p<0.001, $\eta_p^2$ = 0.198; group × trial interaction: $F_{(9.368,505.856)}$ = 0.994, p=0.445, $\eta_p^2$ = 0.018; Student's unpaired $t$-test on the averaged response, $t_{(58)}$ = 0.162, p=0.872, $\eta_p^2$ < 0.001), and to the US during the conditioning phase ($t_{(58)}$ = 1.210, p=0.231, $\eta_p^2$ = 0.025) (**Figure 4—figure supplement 1**).

Next, we analyzed implicit reactions toward the CS in both conditioning and test sessions. A 2 × 2 mixed ANOVA revealed a nonsignificant group main effect ($F_{(1,58)}$ = 2.952, p=0.091, $\eta_p^2$ = 0.048), a nonsignificant phase main effect ($F_{(1,58)}$ = 2.027, p=0.160, $\eta_p^2$ = 0.034), and a significant group × phase interaction ($F_{(1,58)}$ = 4.705, p=0.034, $\eta_p^2$ = 0.075). CS-related SCRs did not differ between groups during conditioning (p=0.798; Bonferroni corrected) but, during the test, the aPFC group exhibited weaker defensive responses than the OC group (p=0.019; Bonferroni corrected). Unlike the aPFC group, whose implicit reactions to the CS diminished from conditioning to test (p=0.014; Bonferroni corrected), the OC group's responses did not differ in the two phases (p=0.600; Bonferroni corrected) (**Figure 4B and C**).

No significant between-group differences were observed in implicit responses to new tones (2 × 3 mixed ANOVA; main effect of group: $F_{(1,58)}$ = 2.775, p=0.101, $\eta_p^2$ = 0.046; main effect of tone: $F_{(2,116)}$ = 5.857, p=0.004, $\eta_p^2$ = 0.092; group × tone interaction: $F_{(2,116)}$ = 3.739, p=0.027, $\eta_p^2$ = 0.061). While the CS triggered weaker reactions in the aPFC group (p=0.019; Bonferroni corrected), both the $NS_1$ (p=0.203; Bonferroni corrected) and the $NS_2$ (p=0.323; Bonferroni corrected) elicited nonsignificantly different responses in the two conditions. These findings underscored the selectivity of divergent rTMS effects in the aPFC and OC groups specifically for the CS. Fear tuning analysis of the aPFC group's implicit reactions unveiled no differences in SCR amplitudes elicited by the CS and the $NS_1$ (p=0.378; Bonferroni corrected), by the CS and the $NS_2$ (p=1.000; Bonferroni corrected), and by the NSs (p=0.552; Bonferroni corrected). In the case of the OC group, implicit reactions were not different for the CS and the $NS_1$ (p=0.876; Bonferroni corrected) but the $NS_2$ evoked lower SCRs than the CS (p<0.001; Bonferroni corrected) and the $NS_1$ (p=0.041; Bonferroni corrected) (**Figure 4D and E**). Furthermore, no significant group differences were detected in SCRs elicited by $US_2$ during the test session ($t_{(58)}$ = 0.175, p=0.862, $\eta_p^2$ < 0.001) (**Figure 4F**).

The distinctive pattern toward the learned threatening cue persisted during the follow-up session (2 × 2 mixed ANOVA; main effect of group: $F_{(1,58)}$ = 2.141, p=0.149, $\eta_p^2$ = 0.036; main effect of phase: $F_{(1,58)}$ = 26.023, p<0.001, $\eta_p^2$ = 0.310; group × phase interaction: $F_{(1,58)}$ = 3.167, p=0.080, $\eta_p^2$ = 0.052). The aPFC group continued to react more dimly to the CS compared to the OC group (p=0.026; Bonferroni corrected). Both the aPFC (p<0.001; Bonferroni corrected) and the OC (p=0.022; Bonferroni corrected) groups showed decreased responses relative to conditioning (**Figure 4G and H**). Conversely, no significant differences were observed in SCRs evoked by $US_2$ during the follow-up session ($t_{(58)}$ = 0.574, p=0.568, $\eta_p^2$ = 0.006) (**Figure 4I**).

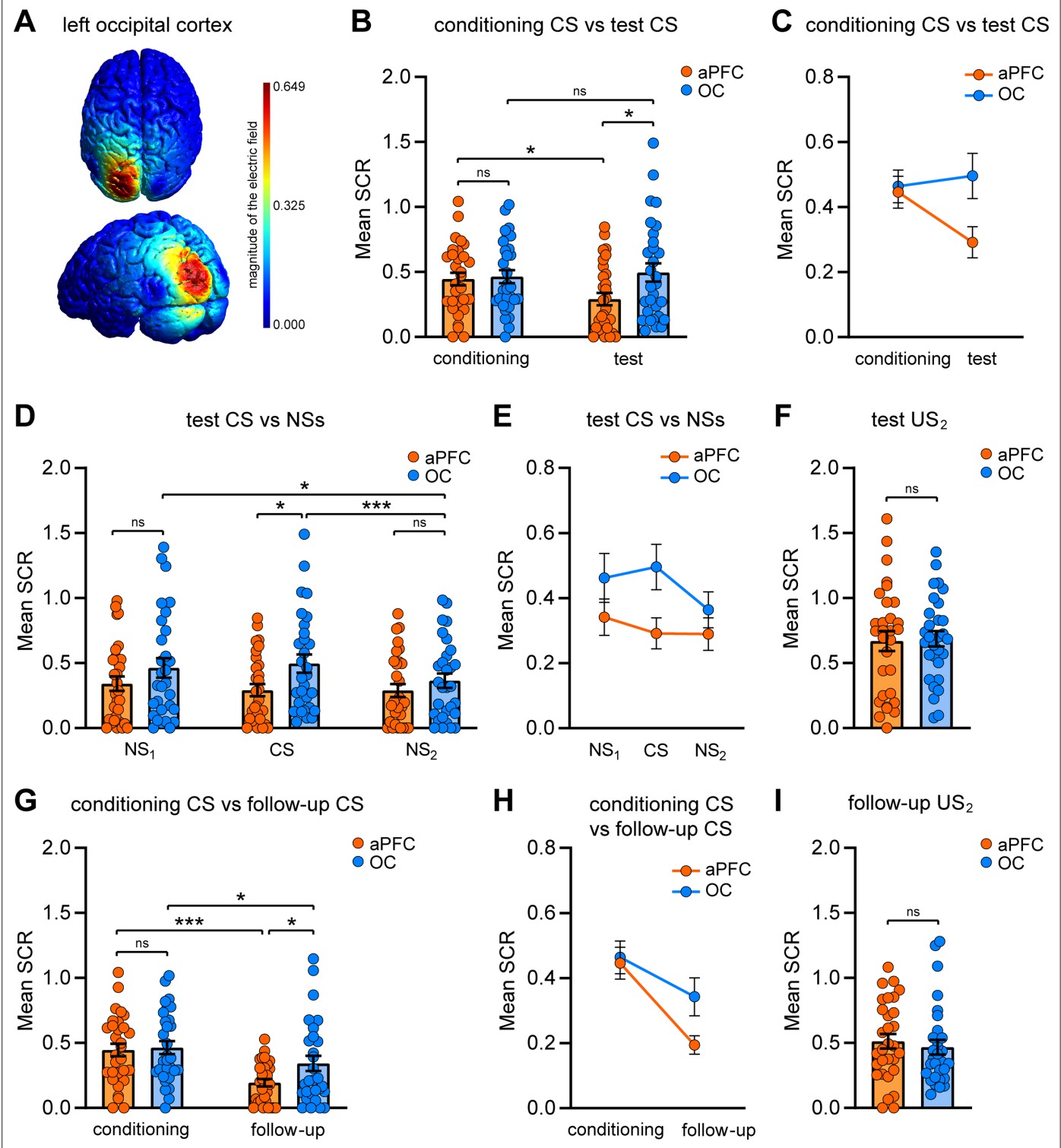

**Figure 4.** Selective effects of repetitive transcranial magnetic stimulation (rTMS) over the anterior prefrontal cortex (aPFC) and the left occipital cortex (OC) on the defensive responses to threat-predictive cues. (**A**) Simulation of rTMS effects on the neural tissue of the left OC (BA 18/19), performed with SimNIBS 4.0 software. The magnitude of the electric field is expressed in V/m. (**B, C**) Dot plot and line chart representing the mean skin conductance responses (SCRs) elicited by the CS during the conditioning session and test session in the OC group (n = 30) compared with the same aPFC group of *Figure 2* (n = 30). The two groups did not differently respond during the conditioning phase, but during the test phase the group stimulated over the aPFC showed weaker reactions than the OC group. While the defensive reactions of the aPFC group decreased from conditioning to test, those of the OC group remained not differently high. (**D, E**) Implicit reactions to NSs during the test session did not differ between groups. In the OC group,

*Figure 4 continued*

the responses elicited by the NS$_2$ were lower than those evoked by the CS and the NS$_1$. (**F**) Implicit reactions to the US$_2$ during the test session were not different between groups. (**G, H**) In the follow-up session, the aPFC group persisted in showing reduced implicit reactions to the CS relative to the OC group. Defensive reactions of both groups decreased from the conditioning phase. Although we found a significant main effect of phase and no group × phase interaction effect, we reported the statistical significance marks of simple main effects. (**I**) Implicit reactions to the US$_2$ during the follow-up session were not different between groups. *p<0.05, ***p<0.001. All data are mean and SEM. 2 × 2 mixed ANOVA followed by Bonferroni-adjusted post hoc comparisons (**B, C, G, H**); 2 × 3 mixed ANOVA followed by Bonferroni-adjusted post hoc comparisons (**D, E**); Student's unpaired *t*-test (**F, I**).

The online version of this article includes the following source data and figure supplement(s) for figure 4:

**Source data 1.** SCR raw data of the OC group during the conditioning, the test, and the follow-up.

**Figure supplement 1.** Implicit reactions during preconditioning (CS) and conditioning (CS, US) in the anterior prefrontal cortex (aPFC) and occipital cortex (OC) groups.

## Comparison between the effects of rTMS administered over the anterior versus the dorsolateral prefrontal cortex

Next, we asked whether the findings we obtained by targeting the aPFC were finely specific for this site or, alternatively, they overlapped with those observed by targeting other prefrontal sub-regions. For this purpose, in one further group (dlPFC, n = 30) we applied the same rTMS procedure over the left dorsolateral PFC (*Figure 5A*) and then compared the implicit patterns of this group with those displayed by the aPFC condition. We selected the left dlPFC since previous studies (e.g., *Raij et al., 2018*) targeted the left hemisphere for testing the rTMS effects on the PFC, and some evidence (see *Marković et al., 2021*) suggested that inhibitory tDCS and rTMS over the left dlPFC may disrupt threat memory consolidation.

We found no significant differences between the two conditions in the post-conditioning US ratings ($t_{(58)}$ = 0.908, p=0.368, $\eta_p^2$ = 0.014) (*Table 1*), in SCRs to the CS during the preconditioning phase ($t_{(58)}$ = 0.967, p=0.337, $\eta_p^2$ = 0.016), to the CS during the conditioning phase (2 × 15 mixed ANOVA; main effect of group: $F_{(1,51)}$ = 0.026, p=0.873, $\eta_p^2$ = 0.001; main effect of trial: $F_{(8.026,409.333)}$ = 12.135, p<0.001, $\eta_p^2$ = 0.192; group × trial interaction: $F_{(8.026,409.333)}$ = 1.042, p=0.403, $\eta_p^2$ = 0.020; Student's unpaired *t*-test on the averaged response, $t_{(58)}$ = 0.378, p=0.707, $\eta_p^2$ = 0.002), and to the US during the conditioning phase ($t_{(58)}$ = 1.752, p=0.085, $\eta_p^2$ = 0.050) (*Figure 5—figure supplement 1*).

Then we compared the implicit reactions toward the CS during conditioning and test sessions. A 2 × 2 mixed ANOVA indicated a nonsignificant main effect of group ($F_{(1,58)}$ = 1.874, p=0.176, $\eta_p^2$ = 0.031), a nonsignificant main effect of phase ($F_{(1,58)}$ = 0.122, p=0.729, $\eta_p^2$ = 0.002), and a significant group × phase interaction ($F_{(1,58)}$ = 10.810, p=0.002, $\eta_p^2$ = 0.157). CS-evoked SCRs did not differ between the two groups during conditioning (p=0.647; Bonferroni corrected) while during the test we found weaker defensive responses in the aPFC group relative to the dlPFC group (p=0.009; Bonferroni corrected). At odds with the aPFC group whose implicit reactions to the CS were diminished from conditioning to test (p=0.013; Bonferroni corrected), the dlPFC group increasingly responded during the test relative to conditioning (p=0.042; Bonferroni corrected) (*Figure 5B and C*). This incremental trend is in line with a previous study that delivered a 1 Hz-rTMS protocol over the left dlPFC (*Borgomaneri et al., 2020*).

We found no between-group differences in the implicit responses to the new tones (2 × 3 mixed ANOVA; main effect of group: $F_{(1,58)}$ = 3.967, p=0.051, $\eta_p^2$ = 0.064; main effect of tone: $F_{(2,116)}$ = 2.819, p=0.064, $\eta_p^2$ = 0.046; group × tone interaction: $F_{(2,116)}$ = 3.286, p=0.041, $\eta_p^2$ = 0.054) since to both the NS$_1$ (p=0.188; Bonferroni corrected) and the NS$_2$ (p=0.110; Bonferroni corrected) were not significantly different. These data showed that the divergent rTMS effects in the aPFC and the dlPFC groups were selective for the CS. Fear tuning analysis of the dlPFC group's implicit reactions revealed no different SCR amplitudes elicited by the CS and the NS$_1$ (p=0.158; Bonferroni corrected) and by the NSs (p=0.721; Bonferroni corrected), but the NS$_2$ evoked lower SCRs than the CS (p=0.014; Bonferroni corrected) (*Figure 5D and E*). We also detected no significant differences between groups in the SCRs elicited by the US$_2$ during the test session ($t_{(58)}$ = 1.762, p=0.083, $\eta_p^2$ = 0.051) (*Figure 5F*).

The different pattern toward the learned threatening cue was replicated during the follow-up session (2 × 2 mixed ANOVA; main effect of group: $F_{(1,58)}$ = 3.751, p=0.058, $\eta_p^2$ = 0.061; main effect of phase: $F_{(1,58)}$ = 3.114, p=0.083, $\eta_p^2$ = 0.051; group × phase interaction: $F_{(1,58)}$ = 15.248, p<0.001, $\eta_p^2$ = 0.208) since the aPFC group persisted in more dimly reacting to the CS with respect to the

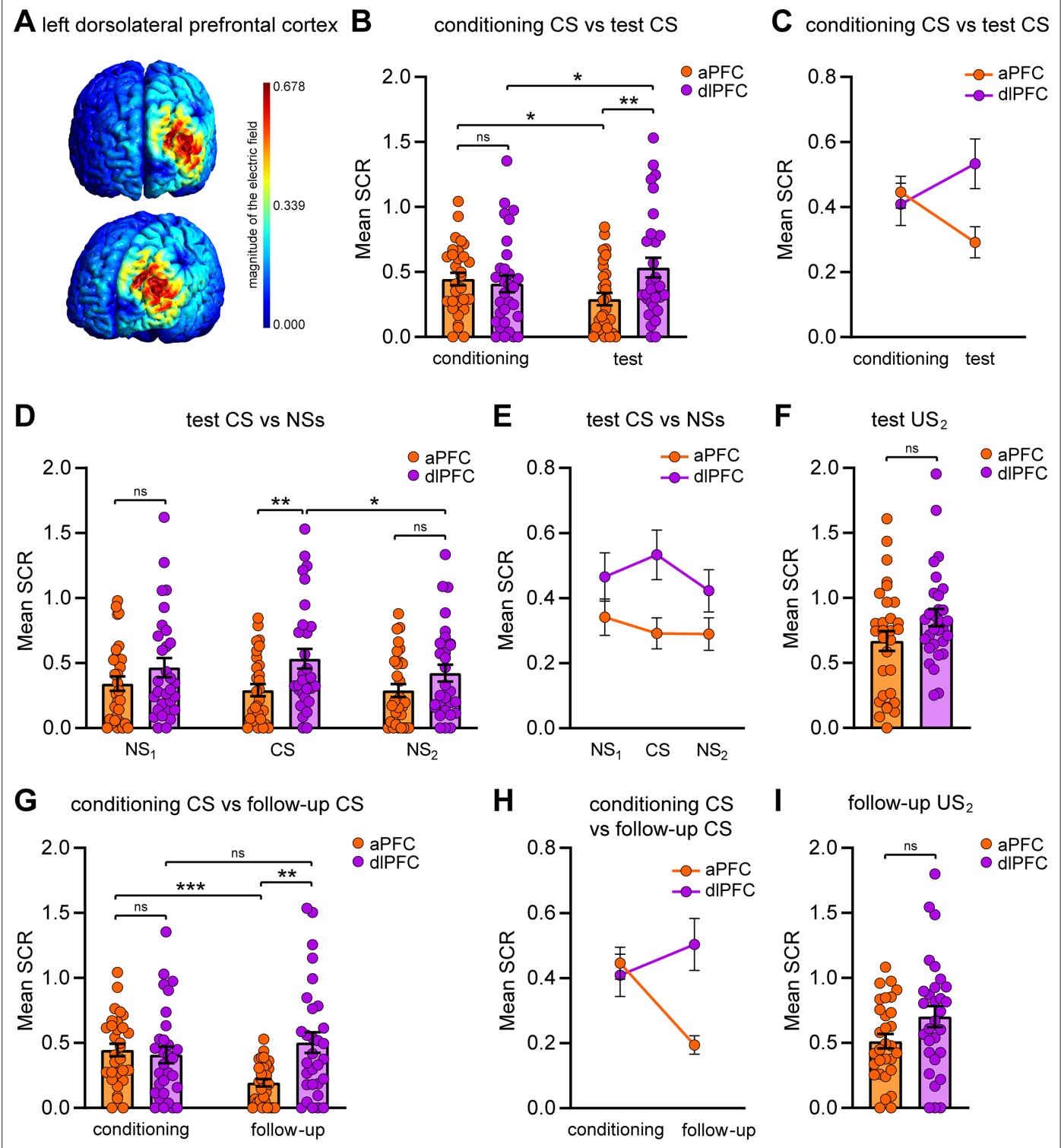

**Figure 5.** Different effects of repetitive transcranial magnetic stimulation (rTMS) over the anterior prefrontal cortex (aPFC) and the left dorsolateral prefrontal cortex (dlPFC) on immediate and remote implicit threat memory. (**A**) Simulation of rTMS effects on the neural tissue of the left dlPFC (BA 8/9), performed with SimNIBS 4.0 software. The magnitude of the electric field is expressed in V/m. (**B, C**) Dot plot and line chart representing the mean skin conductance responses (SCRs) elicited by the CS during the conditioning session and test session in the dlPFC group (n = 30) compared with the same aPFC group of **Figure 2** (n = 30). The two conditions did not differently react during the conditioning phase, whereas during the test phase the group stimulated over the aPFC displayed lower reactions than the dlPFC group. Implicit reactions of the aPFC group decreased from conditioning to test, while those of the dlPFC group increased. (**D, E**) Implicit reactions to NSs during the test session did not differ between groups. In the dlPFC group, the

*Figure 5 continued on next page*

*Figure 5 continued*

responses elicited by the NS₂ were lower than those evoked by the CS. (**F**) The two groups did not differently react to the US₂ during the test session. (**G, H**) In the follow-up session, the aPFC group persisted in more dimly reacting to the CS relative to the dlPFC group and to the conditioning phase. (**I**) Implicit reactions to the US₂ during the follow-up session were not different between groups. *p<0.05, **p<0.01, ***p<0.001. All data are mean and SEM. 2 × 2 mixed ANOVA followed by Bonferroni-adjusted post hoc comparisons (**B, C, G, H**); 2 × 3 mixed ANOVA followed by Bonferroni-adjusted post hoc comparisons (**D, E**); Student's unpaired *t*-test (**F, I**).

The online version of this article includes the following source data and figure supplement(s) for figure 5:

**Source data 1.** SCR raw data of the dlPFC group during the conditioning, the test, and the follow-up.

**Figure supplement 1.** Implicit reactions during preconditioning (CS) and conditioning (CS, US) in the anterior prefrontal cortex (aPFC) and dorsolateral prefrontal cortex (dlPFC) groups.

dlPFC group (p=0.001; Bonferroni corrected), and the aPFC group endured in displaying attenuated responses relative to conditioning (p<0.001; Bonferroni corrected) while the dlPFC group did not (p=0.136; Bonferroni corrected) (*Figure 5G and H*). No significant differences were instead observed in SCRs evoked by the US₂ during the follow-up session ($t_{(58)}$ = 1.927, p=0.059, $\eta_p^2$ = 0.060) (*Figure 5I*).

These findings demonstrated that rTMS over the left dorsolateral PFC did not diminish implicit defensive reactions in the absence of an extinction paradigm, as in other previous studies (*Guhn et al., 2014*; *Raij et al., 2018*; *Borgomaneri et al., 2020*). Meanwhile, rTMS targeting the aPFC proved to be effective in achieving this outcome.

## Discussion

In this study, we found that implicit reactions to both learned and novel stimuli were significantly downregulated following a 1 Hz-rTMS procedure over the aPFC.

So far, most rTMS studies targeting the prefrontal cortex have been conducted to enhance fear extinction processes. A study (*Guhn et al., 2014*) administering one session of 10 Hz-rTMS over the mPFC observed enhancement of extinction learning. These behavioral results were mirrored by the functional near-infrared spectroscopy (fNIRS) findings, which revealed increased mPFC activity in the stimulated group relative to the sham group (*Guhn et al., 2014*). Subsequently, *Raij et al., 2018* delivered brief 20 Hz-rTMS trains over the left posterior PFC – a region that showed robust functional connectivity with the vmPFC– during extinction learning and found a reduction of defensive responses during extinction recall.

Our study differs from the previous ones because we tested rTMS effects over the medial aPFC (medial BA 10), and we did not include extinction training before retrieval. We observed a significant decrease in defensive reactions shortly after rTMS, and this effect was maintained until the follow-up session. Thus, we identified a previously unexplored prefrontal region, the modulation of which can efficiently and durably inhibit implicit reactions to learned threats. These dampening effects may be due to the fact that rTMS over the aPFC have directly modulated the defensive responses activated by the implicit threat memory trace. Alternatively, the rTMS procedure over the aPFC may have inhibited the recall of the CS-US association, preventing the defensive responses from being activated by the CS. This possibility would be in line with a large body of literature on humans (see *Hiser and Koenigs, 2018*), which demonstrates the importance of the medial PFC for value-based processing.

Autonomic reactions to the new tones in the aPFC group relative to the sham control group did not support the conclusion that rTMS targeted threat generalization, leaving open the question of the specificity of rTMS effects. However, the lack of between-group differences in the autonomic responses to the US₂ seems to suggest that the observed effect may be memory-related and not due to a general dampening of autonomic reactivity. Interestingly, defensive responses toward the NSs were decreased following the stimulation of the left occipital cortex (OC group, BA18/19). This effect might be explained by the fact that anatomical and functional reciprocal projections between the medial BA10 and visual association cortices (including BA17/18/19) have been traced via the fronto-occipital fasciculus (FOF) of the human brain (*Catani et al., 2012*; *Orr et al., 2015*; *Wu et al., 2016*; but see *Peng et al., 2018*).

Regarding the persistence of inhibitory effects during the follow-up session, different factors may have contributed to this result. Firstly, the inhibition of SCR responses induced by rTMS during the mnemonic retention test could have persistently reduced such conditioned responses even at a

distance from the treatment. Moreover, the inhibition of these responses during the test might have boosted the extinction of these responses, contributing to keeping them low over time. On this possibility, it should be pointed out that one core knowledge about extinction is that under certain circumstances – such as a simple passage of time (i.e., spontaneous recovery) or a change in surrounding context (i.e., renewal) – extinguished reactions triggered by the CS may reoccur, giving rise to the phenomenon known as return of fear (*Vervliet et al., 2013*; *Pavlov, 1927*; *Rachman, 1966*). To test potential renewal phenomena, which have not been investigated in the aforementioned studies (*Guhn et al., 2014*; *Raij et al., 2018*), we opted for a context-shift amongst the learning (context A), the test (context B), and the follow-up phase (context A), and we found downregulated defensive reactions in both the test and the follow-up phases. These data demonstrated that the aPFC-rTMS protocol long-term reduced threat memory expression in a different context as well as in the context in which the threatening experience had occurred, thus preventing the return of fear. Finally, we cannot exclude that the rTMS applied immediately before the mnemonic retention test interfered with the reconsolidation process that is known to occur after this test (*Su et al., 2022*), resulting in a persistent impairment in the retention of this mnemonic trace.

To potentiate the neural activity of the PFC, both the aforementioned studies (*Guhn et al., 2014*; *Raij et al., 2018*) adopted high-frequency rTMS protocols, which are conventionally considered excitatory of proximal brain activity (*Pascual-Leone et al., 1998*). In our study, we adopted a low-frequency rTMS protocol, which is conventionally considered inhibitory (*Pascual-Leone et al., 1998*). Recent evidence, however, challenged this common frequency-dependent rule (*Luber and Deng, 2016*). Resting-state functional MRI studies demonstrated that 1 Hz-rTMS protocols may also induce downstream distal effects and enhance functional connectivity amongst the brain regions located underneath the coil and remote brain areas of the stimulated neural network (*Beynel et al., 2020*). Additionally, some studies (*Eisenegger et al., 2008*; *Nahas et al., 2001*) reported that 1 Hz-rTMS procedures delivered over the PFC may paradoxically increase regional cerebral blood flow.

The dorsolateral PFC is another prefrontal region that is assumed to be critically involved in threat learning (*Dunsmoor et al., 2007*; *Dunsmoor et al., 2008*) and the downregulation of the cortico-meso-limbic network (*Vicario et al., 2019*). One investigation (*Borgomaneri et al., 2020*) probed the effects of a 1 Hz-rTMS over the dlPFC after memory reactivation to disrupt threat-memory reconsolidation. Stimulated groups failed to discriminate between threatening and safe stimuli, with an increase in autonomic responses to these last ones. A more recent study (*Su et al., 2022*) adopted the continuous theta-burst stimulation (cTBS) over the right dlPFC during the reconsolidation window and successfully decreased the defensive responses for threat memories. In our study, we found an immediate and long-term reduction of defensive responses to the CS only in subjects that were stimulated over the aPFC, while reactions to the NSs were decreased in both conditions. This evidence suggests that targeting the aPFC might represent a more promising approach for therapeutic applications. The lack of any downregulation of CS-evoked reactions that we found in the dlPFC group, at odds with previous studies targeting the same cortical area (*Borgomaneri et al., 2020*; *Su et al., 2022*), might be due either to the fact that we did not adopt an extinction paradigm or to the different brain stimulation approach (rTMS vs. cTBS).

The neural mechanisms by which rTMS over the aPFC decreases threat-conditioned responses can be manifold. Fear memories are formed and retrieved by an intricate neural network encompassing the amygdala (*Pape and Pare, 2010*), the cerebellum (*Sacchetti et al., 2002*; *Sacchetti et al., 2004*; *Zhu et al., 2011*), and sensory cortices (*Sacco and Sacchetti, 2010*; *Grosso et al., 2015*; *Manassero et al., 2018*; *Cambiaghi et al., 2016a*; *Cambiaghi et al., 2016b*; *Concina et al., 2019*; *You et al., 2021*; *You et al., 2022*; *Ojala et al., 2022*; *Monfils, 2022*). Indeed, previous evidence showed both structural connections between the aPFC and the amygdala (*Bramson et al., 2020*; *Folloni et al., 2019*; *Peng et al., 2018*) and a connectivity pathway of downstream modulation from the aPFC to the vmPFC (*Fonzo et al., 2017b*). This projection is activated during fear regulation (*Klumpers et al., 2010*), possibly supporting the vmPFC in top-down modulating the amygdala (*Motzkin et al., 2015*). Through the direct or indirect connections of the aPFC with these areas, it might be that the effects of focal manipulations of aPFC activity reflect complex and dynamic changes in the overall neural network state and/or influence the activity of some of these areas.

Although previous studies enlightened the role of the medial BA10 and BA10–posterior hippocampus functional connectivity in episodic memory retrieval (see *Faran, 2023*), we did not detect any

rTMS-driven effect on explicit recognition memory. The observed divergence between autonomous and declarative patterns might have been due to a selective rTMS action upon the neural system supporting implicit threat processing, which has been widely dissociated from the neural system underlying explicit memory processes (*Bechara et al., 1995*; *LaBar and Cabeza, 2006*; *Knight et al., 2009*). Critically, an rTMS procedure that shapes implicit overreactions to learned threats without affecting conscious knowledge of danger might represent a strategic advantage for therapeutic applications.

Since prevention of relapse is the main challenge for therapies dedicated to post-traumatic and anxiety disorders, our findings may represent an advance in this direction by providing a potential strategy to deactivate emotional overreactions and, most of all, to prevent the return of fear. Future research perspectives might consist of exploring this rTMS application over the aPFC in clinical populations displaying high levels of anxiety or suffering from anxiety disorders and PTSDs.

## Materials and methods

### Participants

All participants (n = 183) were healthy volunteers (mean age: 23.86 ± 2.90, 74 males and 109 females) with no history of psychiatric disorders, neurological illnesses, cardiovascular diseases, illegal drug use, musical training, or any other exclusion criteria for rTMS administration (*Rossi et al., 2021*). During the pre-experimental screening phase, each volunteer was also administered the *State-Trait Anxiety Inventory Form Y* (*Spielberger et al., 1983*; *Pedrabissi and Santinello, 1989*), and those who showed a score >80 in the sum of the two subscales (State + Trait anxiety) were not included in the sample (see *Table 1* for all groups' mean State-Trait Anxiety Inventory scores). Participants were then randomly assigned to each experimental condition, based on sex and age (see *Table 1* for all groups' mean age and sex distribution). We discarded 11 participants because of a complete absence of SCRs during the test session, leaving a total of 172 participants. Each participant provided written informed consent after receiving a complete description of the experimental procedures. All experimental procedures were performed in accordance with the ethical standards of the Declaration of Helsinki and were approved by the Bioethics Committee of the University of Turin (protocols nos. 19961 and 161427).

### Auditory stimuli

Auditory stimuli were pure sine wave tones with oscillation frequencies of 800 Hz (CS), 1000 Hz ($NS_1$), and 600 Hz ($NS_2$), lasting 6 s with onset/offset ramps of 5 ms. Tones were digitally generated using Audacity 2.1.2 software (Audacity freeware). The unconditioned threatening stimulus ($US_2$) consisted of a woman scream sample lasting 4 s. All auditory stimuli were binaurally delivered through headphone speakers (Direct Sound EX29) at 50 dB intensity. All experimental scenarios were controlled by Presentation 21.1 software (NeuroBehavioral Systems, Berkeley, CA).

### Preconditioning

This phase consisted of the presentation of four trials of the CS (800 Hz) with an inter-trial interval (ITI) randomly ranging between 21 s and 27 s. SCRs were recorded during this phase to provide a baseline response pattern to the 800 Hz tone for each participant. At the end of this phase, participants were asked to confirm whether the tones were easily audible but not too loud or annoying.

### Unconditioned stimulus calibration procedure

Before starting with the calibration procedure, systolic and diastolic blood pressure was measured to prevent possible hypo-arousal reactions caused by basal hypotension. The US consisted of a mild electrical shock (train pulse at 50 Hz lasting 200 ms, with a single pulse duration of 1000 μs) generated with a DC stimulator (DS7A Constant Current Stimulator, Digitimer). Impulses were delivered through a bar stimulating electrode connected by a Velcro strap on the upper surface of the dominant hand's index finger. The electrical stimulation intensity was individually calibrated through a staircase procedure (*Manassero et al., 2019*; *Manassero et al., 2022*; *Cornsweet, 1962*), starting with a low current near the perceptible tactile threshold (~0.5 mA). Participants were asked to rate the painfulness of each train pulse on a scale ranging from 0 (not painful at all), 1 (pain threshold) to 10 (highly painful if

protracted in time). At the end of the procedure, the US amplitude was then set at the current level (mA) corresponding to the mean rating of '7' on the subjective analog scale.

## Conditioning

After a 1 min resting period, participants underwent a single-cue auditory threat conditioning, which consisted of the presentation of 15 trials of the CS (CS, 800 Hz), with an ITI randomly ranging between 21 s and 27 s. The CS co-terminated with the US 12 times (80% reinforcement rate). Subjects were not informed about any possible CS-US contingency. To validate the threat learning experience, immediately following this phase subjects rated the painfulness of the US using the same analog scale as in the preconditioning calibration procedure (see *Table 1* for all groups' US current intensity and US analog ratings).

## Transcranial magnetic stimulation

Transcranial magnetic stimulation was performed with a Magstim Rapid (*Wilker et al., 2014*) Stimulator (Magstim Co., Whitland, Dyfed, UK). A 70 mm figure-of-eight coil was positioned over the subject's M1 cortical area at the optimum scalp position to elicit a contraction of the contralateral abductor *pollicis brevis* muscle. Resting motor threshold (rMT) was defined as the minimum stimulation intensity that induced a visible finger movement in at least 5 out of 10 single pulses over the right-hand area of the left primary motor cortex (*Guhn et al., 2014*; *Westin et al., 2014*). After having determined each individual's rMT, we applied a single train of 1 Hz-rTMS (*Ando et al., 2015*; *Salatino et al., 2019*) for a total duration of 10 min (600 pulses) to the target area. The rTMS intensity was set at 80% of the rMT for subjects whose rMT was ≤50% of the machine's maximum deliverable power (e.g., the intensity corresponded to 40% of the maximum power when the rMT was equal to 50% of the same parameter). For subjects with an rMT > 50%, the stimulation intensity was always set to a ceiling corresponding to 40% of the machine's maximum deliverable power (see *Table 1* for each group's mean rMT and mean stimulation intensity). During the rTMS procedure, participants were seated in a comfortable recliner that we adjusted to allow their upper body to be in a sloped position, thus ensuring an optimal positioning of the coil.

To target the medial anterior portion of the prefrontal cortex (BA 10; aPFC and aPFC-E groups), the coil was centered over Fpz (10% of nasion-inion distance) according to the international 10–20 EEG system (*Jasper, 1958*; *Figure 1*). This placement should – with an rTMS reach of 1.5–2 cm beneath the scalp (*Epstein et al., 1990*; *Rudiak and Marg, 1994*) – ensure the targeting of the medial aPFC as in previous studies (*Guhn et al., 2014*; *Herrmann et al., 2017*; *Karmann et al., 2016*) and avoid the targeting of the dmPFC, which would have been localizable with a scalp-based heuristic approach of 25.84% nasion-inion distance (*Mir-Moghtadaei et al., 2016*). In the case of left occipital cortex stimulation (OC group), the coil was positioned over O1 using the 10–20 EEG system (BA 18/19), which functionally corresponds to associative visual cortices V3, V4, and V5 (*Rojas et al., 2018*; *Brighina et al., 2003*; *Figure 1*). For the stimulation of the left dorsolateral prefrontal cortex (dlPFC group), the coil was placed over F3 using the 10–20 EEG system (BA 8/9) (*Borgomaneri et al., 2020*; *Mir-Moghtadaei et al., 2015*; *Figure 1*). For sham stimulation (sham and sham-E groups), the coil was centered over Fpz and positioned perpendicular to the scalp surface, so that no effective stimulation reached the brain during the procedure but allowed subjects to feel a comparable coil-scalp contact and hear the same noise as in real stimulation (*Figure 1*).

All participants were blinded to their experimental condition (i.e., active or sham) and were not informed about the potential cognitive or emotional effects of the stimulation.

## Discomfort-inducing procedure

The discomfort-inducing procedure mirrored the rTMS protocol and consisted of the delivery of mild electrical shocks (single 1 Hz train of 600 pulses lasting 10 min, with a single pulse duration of 500 μs to mimic the duration of a single TMS pulse) generated with a DC stimulator (DS7A Constant Current Stimulator, Digitimer). Impulses were delivered through two cup-stimulating electrodes attached to the surface of the subject's forehead in correspondence with Fpz according to the 10–20 EEG system. As in the case of the US calibration, the electrical stimulation intensity was individually calculated through a staircase procedure (*Cornsweet, 1962*), starting with a low current near the perceptible tactile threshold (~0.5 mA). Participants were asked to evaluate the perceived discomfort of each

pulse on a scale from 0 (no discomfort) to 10 (high discomfort). At the end of the procedure, the shock amplitude was set at the current level (mA) corresponding to the mean rating of '4' on the subjective analog scale. To quantify the habituation to the uncomfortable stimulations, at the end of every minute of the 10 min procedure (i.e., every 60 pulses), subjects were requested to rate the level of the present discomfort on the same scale adopted during the calibration procedure.

### Implicit recognition test

After a 1 min resting period, participants underwent this task, which consisted of the presentation of 12 auditory stimuli in a completely random sequence: $4 \times CS$, $4 \times NS_1$, $4 \times NS_2$, with an ITI whose duration randomly ranged between 21 s and 27 s. SCRs were recorded throughout this phase, and the stimulating electrode was kept attached to create the expectation of receiving the US (*Ameli et al., 2001*). Differently from other paradigms (*Onat and Büchel, 2015*; *Lissek et al., 2014*; *Dunsmoor et al., 2017*; *Holt et al., 2014*), here no shocks were delivered to avoid any reacquisition effect (*Manassero et al., 2019*; *Manassero et al., 2022*).

### Implicit unconditioned threatening test

This task was designed to elicit an unconditioned electrodermal response and consisted of the presentation of four trials of a woman scream sample lasting 4 s, with an ITI randomly ranging between 21 s and 27 s. SCRs were recorded throughout this phase, and the stimulating electrode was kept attached.

### 2AFC explicit recognition test

This procedure involves the presentation of two stimuli on each trial and the subject chooses the one that was previously encoded (i.e., the first or the second one). As in our previous works (*Manassero et al., 2019*; *Manassero et al., 2022*), a 2AFC design was preferred over a new-old paradigm, which involves one single stimulus on each trial, and the subject judges whether the stimulus has been previously encoded (old), or whether it is new. Our choice was motivated by the evidence that a 2AFC task improves recognition performance and discourages response biases such as the familiarity-based decision bias, namely the heuristic to endorse novel cues as 'old' when their familiarity is high (*Macmillan and Creelman, 2004*).

The task consisted of the presentation of 16 tone-pairs, each composed of the CS (800 Hz) and one of the two NSs ($NS_1$, 1000 Hz or $NS_2$, 600 Hz) in a completely random sequence: $4 \times CS$ vs. $NS_1$, $4 \times NS_1$ vs. CS, $4 \times CS$ vs. $NS_2$, $4 \times NS_2$ vs. CS. On each trial, the two stimuli were presented with an intra-trial interval of 1000 ms. After each pair offset, an ITI randomly ranging between 21 s and 27 s occurred. Participants were explained that in each couple of sounds there was a tone that they had heard on the first session (1 week before or, in the case of the follow-up session, 2 weeks before) and a new tone. Participants were then instructed to recognize and verbally report which one (the first or the second) was the tone heard in the first session, paired with the US-shock (CS). Participants were further asked to verbally provide a confidence rating about each response, on a scale from 0 (completely unsure) to 10 (completely sure). No feedback was supplied. As in the implicit task, the stimulating electrode was kept attached, but no shock was delivered.

### 2AFC perceptual discrimination test

The task consisted of the presentation of seven pairs of auditory stimuli (i.e., CS vs. $NS_1$, $NS_1$ vs. CS, CS vs. $NS_2$, $NS_2$ vs. CS, CS vs. CS, $NS_1$ vs. $NS_1$, $NS_2$ vs. $NS_2$) with a 1000 ms intra-pair interval in a completely random sequence (ITI randomly ranging between 21 s and 27 s). For each pair, subjects were asked to report whether the two tones were 'the same tone or different tones' and to provide a confidence rating on an analog scale from 0 (completely unsure) to 10 (completely sure). No feedback was supplied, and the stimulating electrode was kept attached.

### Psychophysiological recording and analysis

Event-related SCRs were used as an implicit index of defensive responses. To record the autonomic signal, two Ag-AgCl non-polarizable electrodes filled with isotonic paste were attached to the index and middle fingers of the non-dominant hand by Velcro straps. The transducers were connected to the GSR100C module of the BIOPAC MP-150 system (BIOPAC Systems, Goleta, CA) and signals were recorded at a channel sampling rate of 1000 Hz. SCR waveforms were analyzed offline using

AcqKnowledge 4.1 software (BIOPAC Systems) and performed blindly to the subject's experimental condition and the randomized sequence of stimuli. Each SCR was evaluated as event-related if the trough-to-peak deflection occurred 1–6 s (for the CS and the NSs) or 1–4 s (for the $US_2$) after the stimulus onset, the duration was comprised between 0.5 and 5.0 s, and the amplitude was greater than 0.02 µS. Responses that did not fit these criteria were scored zero. To account for inter-individual variability, these raw values were then scaled according to each participant's average unconditioned response by dividing each response by the mean US response during the conditioning phase (*Schiller et al., 2010*; *Battaglia et al., 2018*). Scaled SCR data were square-root transformed to normalize the distributions (*Lykken and Venables, 1971*).

## Statistical analyses

We computed the appropriate sample size based on a power analysis performed through G*Power 3.1.9.2. For the main statistics, that is, mixed ANOVA (within–between interaction) with two groups and two measurements, with the following input parameters: α = 0.05, power (1-β) = 0.95, and a hypothesized effect size (f) = 0.25, the estimated sample size resulted in n = 30 per experimental group.

Since most variables passed the D'Agostino–Pearson omnibus normality test, parametric statistics were adopted in each experiment.

To test the between-group differences in post-conditioning US ratings, preconditioning mean SCRs levels, mean SCRs to the CS and the US during conditioning, and mean SCRs to the $US_2$ during the test and the follow-up sessions, we performed Student's unpaired *t*-tests. Potential differences in CS-related SCRs over the 15 trials of the conditioning phase were tested through 2 × 15 mixed ANOVAs with group (aPFC vs. sham, aPFC vs. OC, aPFC vs. dlPFC) as between-subject variable and trial (1–15) as within-subject variable.

To test the potential between-group differences in the implicit reactions to the CS during the conditioning session, the test session, and the follow-up session, as well as the within-group differences from conditioning to test/follow-up phases, we computed 2 × 2 mixed ANOVAs with group (aPFC vs. sham, aPFC vs. OC, aPFC vs. dlPFC, sham vs. ctrl discomfort) as between-subject variable and phase (conditioning vs. test, conditioning vs. follow-up) as within-subject variable. Bonferroni adjustment was applied for simple main effects analyses. To compare between-group and within-group responses to the CS and the NSs during the test session, we performed 2 × 3 mixed ANOVAs with group (aPFC vs. sham, aPFC vs. OC, aPFC vs. dlPFC) as between-subject variable and tone ($NS_1$, CS, and $NS_2$) as within-subject variable. Bonferroni adjustment was applied for simple main effects analyses.

To test the between-group differences in the explicit recognition and respective confidence ratings, as well as in the perceptual discrimination and respective confidence ratings during the test and the follow-up sessions (aPFC-E vs. sham-E), we performed Student's unpaired *t*-tests. To test whether explicit recognition levels were significantly higher than the 50% chance level for each condition during the test and the follow-up sessions, we calculated Student's one-sample *t*-tests against 0.50.

For each ANOVA, we assessed the sphericity assumption through Mauchly's test. Where it was violated, we applied the Greenhouse–Geisser correction accordingly.

The null hypothesis was rejected at p<0.05 significance level. All statistical analyses were performed using SPSS Statistics 22 (IBM) and Prism 9 (GraphPad).

## Acknowledgements

We thank all the subjects for their participation in this study. We also thank Melania Lattuada, Veronica Cintori, Ester Fusaro, Alessia Di Blasi, and Adriana Monno for their help in the experimental procedures and for their continuous support and advice. This work was supported by the 'Compagnia di San Paolo, Progetto d'Ateneo', University of Turin 2017 (CSTO167503), the Fondazione Giovanni Goria and Fondazione CRT (Talenti della Società Civile, ed. 2018), the Grant 'Progetti di ricerca di Rilevante Interesse Nazionale (PRIN)' 2017 (project no. 20178NNRCR_002) from the Italian Ministry of University and Research (MIUR), Fondazione Cariverona 2020, Fondazione CRT 2021, and Banca D'Italia 2023.

# Additional information

## Funding

| Funder | Grant reference number | Author |
| --- | --- | --- |
| Compagnia di San Paolo | CSTO167503 | Benedetto Sacchetti |
| Fondazione Giovanni Goria | | Eugenio Manassero |
| Fondazione CRT | | Benedetto Sacchetti |
| Banca d'Italia | | Benedetto Sacchetti |
| Fondazione Cassa di Risparmio di Verona Vicenza Belluno e Ancona | | Benedetto Sacchetti |
| Italian Ministry of University and Research | | Benedetto Sacchetti |

The funders had no role in study design, data collection and interpretation, or the decision to submit the work for publication.

## Author contributions

Eugenio Manassero, Conceptualization, Formal analysis, Investigation, Methodology, Writing – original draft, Writing – review and editing; Giulia Concina, Maria Clarissa Chantal Caraig, Formal analysis, Investigation; Pietro Sarasso, Adriana Salatino, Investigation; Raffaella Ricci, Supervision, Methodology, Project administration, Writing – review and editing; Benedetto Sacchetti, Conceptualization, Supervision, Funding acquisition, Methodology, Writing – original draft, Project administration, Writing – review and editing

## Author ORCIDs

Adriana Salatino (iD) http://orcid.org/0000-0002-2471-7212
Benedetto Sacchetti (iD) http://orcid.org/0000-0002-8695-8310

## Ethics

Each participant provided written informed consent after receiving a complete description of the experimental procedures. All experimental procedures were approved by the Bioethics Committee of the University of Turin (protocols N. 19961 and N. 161427).

## Decision letter and Author response

Decision letter https://doi.org/10.7554/eLife.85951.sa1
Author response https://doi.org/10.7554/eLife.85951.sa2

# Additional files

## Supplementary files

• MDAR checklist

• Source data 1. SCR raw data of the aPFC group during the conditioning, the test, and the follow-up.

• Source data 2. SCR raw data of the aPFC group, the sham group, the OC group, and the dlPFC group during the pre-conditioning and the conditioning.

## Data availability

All data generated or analysed during this study are included in the manuscript and supporting file.

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
