## [Editor Report]

This study presents a valuable finding that rTMS over the aPFC, applied before threat memory retrieval, can immediately decrease implicit reactions to learned and novel stimuli in humans and that these effects can persist one week later, in the absence of rTMS. The evidence supporting the claims of the authors is solid and raise important hypotheses for further research. The work will be of interest to researchers in the fields of neuromodulation and affective neuroscience.

---

## [Decision Letter]

**Decision letter after peer review:**

Thank you for submitting your article "Medial prefrontal cortex stimulation abolishes implicit reactions to threats and prevents the return of fear" for consideration by *eLife*. Your article has been reviewed by 2 peer reviewers, and the evaluation has been overseen by a Reviewing Editor and Floris de Lange as the Senior Editor. The reviewers have opted to remain anonymous.

*Essential revisions (for the authors):*

General summary

This study presents the useful observation that repetitive Transcranial Magnetic Stimulation over the medial prefrontal cortex exerts dampening effects of conditioned responses and generalization of these responses to similar cues. However, the evidence supporting the claims of the authors is incomplete. If the observations hold with the recommended analyses, the work will be of interest to psychologists and neuroscientists and neuropsychiatrists.

Comments and recommendations:

The reviewers indicate many strengths of the study, including the use of multiple stimulation sites (including sham and active control) and the inclusion of a follow-up test and a generalization tests. However, several conceptual, methodological and interpretational issues remain that currently limit the value of potential conclusions drawn from this work. There are a number issues that have to be addressed, those include an alternative analysis approach (rev 1 and 2), issues related to the stimulation sites, and the effects of potential sensory confounds (rev 2). I have summarized the most important issues underneath. There were also a number of specific recommendations by each of the reviewers that we would like you to take into account (see individual review reports for those suggestions).

1) Pre-processing should be in line with accepted standards in the fear conditioning field: Start out to range correct each individual SCR datapoint (and clarify what exact value (e.g., average, maximum) of the US during acquisition is used to do so). Differences scores with a preconditioning phase are very uncommon, and also not necessary when a comparison with the acquisition data is being made in the main analyses. No matter what, only as a final step, should the data be square root transformed.

2) Critical analyses are lacking, namely within subject analyses. Thus, the main analyses should incorporate a within-subject factor of Time (pre and post intervention), possibly also with the within- subject factor Trial Number (e.g., the last 4 trials of acquisition and the four test trials) such that an omnibus test can be performed to assess whether there is an interaction with Time and Group (and or Trial Number). If so, this can be followed-up with relevant planned contrasts investigating whether fear is significantly reduced in the mPFC group, and whether the DIFFERENCE in fear reduction is higher in the mPFC group as compared to such difference scores in the other groups.

This also counts for the follow-up analyses: there should be within-subject comparisons.

3) The generalization analyses should consist of the within subject factor Stimulus Type (CS, NS1, NS2) and between subject factor Group. Again, only in the case of a significant omnibus test, these should be followed up by relevant planned comparisons.

4) Even if the pattern of results holds, then the claim that the long- term follow-up reductions of fear were achieved in the absence of any extinction cannot be made with confidence: after all, upon mPFC stimulation during the second session, the CS was presented four times, and so were each of the two generalization stimuli. So perhaps extinction was not complete, but almost certainly some extinction has taken place: it is well-known that the strongest extinction-learning typically takes place in the first trials (e.g., due to higher prediction errors). The authors do not give any alternative theoretical explanation for the enduring reduction of fear reduction, which would be interesting to learn their thoughts on.

5) There is a discrepancy between the background in the introduction and the eventual aim to stimulate BA10: In the introduction the authors sketch that what they call 'mPFC' is involved in regulation of threat responses. The authors make a convincing case, however, almost all of the evidence they present concerns the ventromedial part of the prefrontal cortex (refs 18-25). Then they mention they aim to stimulate BA10, however, BA10 is anatomically, cytoarchitectonically and functionally a completely different area than vmPFC. This raises the question whether the rationale holds given that they stimulate BA10.

The authors should be more precise in their nomenclature throughout the manuscript. In the introduction clarify nomenclature and mention limitations discrepancy of previous literature, ideal aim and operationalization limitations.

6) Although the authors are praised for including both sham and active control regions, the controls might not be optimally chosen to control for the potential confounds of their condition of interest (mPFC-TMS). Namely, TMS on the forehead can be unpleasant, if not painful, whereas sham-TMS or TMS applied to the back of the head or even over dlPFC is not (or less so at the very least). Given that the SCR results after mPFC TMS show exactly the same temporal pattern as the sham-TMS but with a lower starting point, one could wonder whether a painful stimulation prior to the retrieval might already have caused habituation to painful stimulation observed in SCR in consequent CS presentations. A control region that would have been more obvious to take is the lateral part of BA10, by moving the TMS coil several centimeters to the left or right, circumventing all things potentially called medial but giving similar unpleasant sensations (pain etc).

The authors should consider to add simulations of the TMS effects on the neural tissue. That will make it easier for non-experts in TMS to judge the success of your manipulation. In addition, they should discuss in the discussion clearly that one cannot rule out the possibilities that effects may be the consequence of pain in the experimental stimulation condition (and not in control conditions) as this may have 'masked' the 'threat of shock' in the test session.

7) To increase transparency and credibility, ideally, the paper should adopt a complete open science approach: the raw data along with the code to arrive at results should be openly accessible. The acquisition data should also be presented in graphs, both with all trials and the relevant averages that are chosen to compare for the pre-post intervention effect.

8) To further strengthen the theoretical framework of the paper, the rationale to include the explicit memory tests as well as a substantiation of the hypotheses should be included in the introduction. Also, a clear and substantial reasoning behind the inclusion of the generalization tests is required. Currently is seems a bit as if the study was actually set-up as some sort-of reconsolidation intervention. I am saying this because the introduction is oblivious as to per what mechanism could mediate the long-term effects of the intervention, if not extinction of reconsolidation. So a clear reasoning why the long-term test was implemented and why the authors hypothesize what they hypothesize should be added.

9) The discussion could reflect more on what mechanisms may be behind the long-term effect, and how the absence of any effects of the explicit memory tests should be interpreted. A critical reflection on what exact theoretical constructs these tests represent is also required.

To conclude:

There is a possibility that a re-analysis of the data using properly preprocessed SCR data along with analyses that include comparisons with the conditioned responses during acquisition reveal a different pattern of results. Therefore, whether the authors truly achieved their aims and whether the results support their conclusions is as of yet undecided. If the results hold and all of the above issues can be addressed, this study could be a valuable contribution to the field.

*Reviewer #1 (Recommendations for the authors):*

The suggestions that I'd like to present here build on the concerns I have raised in the public review, along with some further suggestions to generally strengthen the manuscript.

– Pre-processing should be in line with accepted standards in the fear conditioning field: Start out to range correct each individual SCR datapoint (and clarify what exact value (e.g., average, maximum) of the US during acquisition is used to do so). Differences scores with a preconditioning phase are very uncommon, and also not necessary when a comparison with the acquisition data is being made in the main analyses. No matter what, only as a final step, should the data be square root transformed.

– The main analyses should incorporate a within-subject factor of Time (pre and post intervention), possibly also with the within-subject factor Trial Number (e.g., the last 4 trials of acquisition and the four test trials) such that an omnibus test can be performed to assess whether there is an interaction with Time and Group (and or Trial Number). If so, this can be followed-up with relevant planned contrasts investigating whether fear is significantly reduced in the mPFC group, and whether the DIFFERENCE in fear reduction is higher in the mPFC group as compared to such difference scores in the other groups.

– The generalization analyses should consist of the within subject factor Stimulus Type (CS, NS1, NS2) and between subject factor Group. Again, only in the case of a significant omnibus test, these should be followed up by relevant planned comparisons.

– Very important: to increase transparency and credibility, ideally, the paper should adopt a complete open science approach: the raw data along with the code to arrive at results should be openly accessible. The acquisition data should also be presented in graphs, both with all trials and the relevant averages that are chosen to compare for the pre-post intervention effect.

– Generally, the factors and their levels of performed ANOVAs should always be reported.

– Pay attention to using consistent numbers after the comma's, this currently varies widely.

– For transparency, always report the exact p-value (so not p>0.05)

– To strengthen the theoretical framework of the paper, the rationale to include the explicit memory tests as well as a substantiation of the hypotheses should be included in the introduction. Also, a clear and substantial reasoning behind the inclusion of the generalization tests is required. It is now after the fact, but preregistration of the hypotheses and analytical approach would have been a major plus. Currently is seems a bit as if the study was actually set-up as some sort-of reconsolidation intervention. I am saying this because the introduction is oblivious as to per what mechanism could mediate the long-term effects of the intervention, if not extinction of reconsolidation. So a clear reasoning why the long-term test was implemented and why the authors hypothesize what they hypothesize should be added. For future studies, preregistration is highly recommended.

– The discussion (which is currently already interesting) could reflect more on what mechanisms may be behind the long-term effect, and how the absence of any effects of the explicit memory tests should be interpreted. A critical reflection on what exact theoretical constructs these tests represent is also required.

*Reviewer #2 (Recommendations for the authors):*

– Labeling all areas in the midline of the prefrontal cortex 'medial prefrontal cortex' might not be wrong, but lumping those very different regions together as a homogeneous functional entity can be a bit off-putting to those who consider anatomy important. Perhaps the authors can be more precise in their nomenclature throughout the manuscript.

– I would suggest to plot the SCR timeseries in addition to the deviation from baseline. That would also give insight into the amount of change caused by the TMS protocol.

– The sentence starting at lines 32-33: "in this scenario…" is quite difficult to understand, perhaps consider revising it.

– Lines 93-95 is quite confusing, for a moment I thought the authors were going to assess the potential distal effects of TMS, which did not happen. Consider keeping this for the discussion

– On several instances the authors equate a non-significant effect to effects being the same e.g. line 109-110: "there the CS evoked similarly strong autonomous reactions (P > 0.05) (Figure 2A)." That is not really what you can conclude from such an analysis as you do not test how similar these effects are. Additionally, it would be good to give the exact p-value and t-statistics for readers to judge how not different these effects are.

– Consider adding simulations of the TMS effects on the neural tissue. That will make it easier for non-experts in TMS to judge the success of your manipulation.

– It is not clear to me why the authors shift back to context A in their third session, is there a rationale for that?

– Why is the time for the peak-to-trough deflection different for US2 as compared to the CS and US?

[Editors' note: further revisions were suggested prior to acceptance, as described below.]

Thank you for resubmitting your work entitled "Medial anterior prefrontal cortex stimulation down-regulates implicit reactions to threats and prevents the return of fear" for further consideration by *eLife*. Your revised article has been evaluated by Floris de Lange (Senior Editor) and a Reviewing Editor.

The manuscript has been improved and the addition of control experiments has many of the concerns raised by reviewer 2 who has no further comments. However, there are some remaining issues related to the SCR data handling and reporting that need to be addressed. Reviewer 1 provides valuable concrete suggestions for analyses (analysing SCR data from all acquisition trials, bonferroni correction where appropriate) and reporting (discussing limitations related to efficacy of threat conditioning, the lack of a control stimulus CS-, failing generalisation manipulation that makes it unwarranted to claim that the intervention targeted generalization). An important possibility that needs to be discussed is that the effects reflect a general dampening effect on seeing any kind of stimulus (that is not aversive in itself).

*Reviewer #1 (Recommendations for the authors):*

The authors have revised the manuscript substantially, and the result certainly has improved. The revised version is clear, nuanced, and the contribution of the manuscript to the literature is evident. Some final considerations and concerns to reflect on to further improve the manuscript.

- In the intro it would be good to add a clear and explicit rationale for the inclusion of the explicit memory test and the generalization test.

– Please also analyse the SCR data from ALL acquisition trials and add its graph on a trial-by-trial basis, this is the best way to assess any possible pre-existing learning differences for the groups.

– One concern I have is the lack of a control stimulus (CS-) during acquisition: as such it is challenging to reveal whether conditioning was effective in the first place. In this light it is also concerning that the SCR values seem very low.

– Please confirm that the shock electrodes were attached during the test phases and indicate so in the manuscript.

– It is unclear what tests were Bonferroni corrected and which ones were not (and why), this should be done consistently.

– The unit of SCR on some of the y-axes is lacking. On some other graphs it is the root of mS. The unit applied should be consistent across all graphs. In the graphs, also the relevant planned comparisons that are NS should be indicated.

– The lack of a main effect of tone in the generalization phase indicates that the generalisation manipulation itself did not work. This needs to be flagged in the relevant analyses and critically discussed in the Discussion section. For one thing, it does not allow the conclusion that the intervention targeted generalization. Consequently, one may wonder whether the effect (at least the immediate one, the follow-up test is another story) is mediated by memory at all, or whether it involves a more general dampening effect on seeing any kind of stimulus (that is not aversive in itself).

---

## [Author Response]

Essential revisions (for the authors):Reviewer #1 (Recommendations for the authors):The suggestions that I'd like to present here build on the concerns I have raised in the public review, along with some further suggestions to generally strengthen the manuscript.– Pre-processing should be in line with accepted standards in the fear conditioning field: Start out to range correct each individual SCR datapoint (and clarify what exact value (e.g., average, maximum) of the US during acquisition is used to do so). Differences scores with a preconditioning phase are very uncommon, and also not necessary when a comparison with the acquisition data is being made in the main analyses. No matter what, only as a final step, should the data be square root transformed.

We would like to thank the Reviewer for raising this important point. To align our data analysis with the accepted standards in the fear conditioning field, we re-analyzed our data starting by range-correcting each participant’s SCR raw data point, dividing it by the same participant’s average US response during the conditioning phase. We finally applied a square-root transformation of each scaled data point (see Methods section).

As usefully suggested, we have also analyzed the responses elicited by the CS during the conditioning phase and we then processed them in the same manner. In line with previous studies in the same field (Raij *et al.,* 2018), the trials in which the US shock was delivered (12 out of 15 trials) were excluded to avoid artifacts/confounds induced by the electric shock itself. Therefore, the remaining 3 out of 15 trials were analyzed. The data we obtained (with a sample of *n* = 21 participants) were not normally distributed, preventing us from running the suggested omnibus ANOVA analysis. Thus, we reasoned to perform a new power analysis and repeat the experiments to reach a sample width of *n* = 30 (for the aPFC ) Throughout the manuscript, we have changed the denomination of the medial prefrontal cortex (mPFC) with the term “medial *anterior* prefrontal cortex” (aPFC, anatomically corresponding to the medial portion of the Brodmann area 10, BA10) in accordance with the suggestion of the Reviewer 2., sham, OC, and dlPFC groups. With this sample width, we reached a normal distribution of the data and we were able to perform the suggested analyses.

– The main analyses should incorporate a within-subject factor of Time (pre and post intervention), possibly also with the within-subject factor Trial Number (e.g., the last 4 trials of acquisition and the four test trials) such that an omnibus test can be performed to assess whether there is an interaction with Time and Group (and or Trial Number). If so, this can be followed-up with relevant planned contrasts investigating whether fear is significantly reduced in the mPFC group, and whether the DIFFERENCE in fear reduction is higher in the mPFC group as compared to such difference scores in the other groups.

Thank you for this useful suggestion. After preprocessing the data as indicated in the previous point, we ran omnibus 2 × 2 mixed ANOVAs by incorporating a within-subject factor of Time/Phase (pre- and post-intervention) both in the case of the test (i.e., conditioning *vs* test) and the follow-up (i.e., conditioning *vs* follow-up). Therefore, we included in the omnibus analysis the averaged SCRs level evoked by the CS during these phases. These analyses allowed us to assess whether there was an interaction with Time/Phase and Group, and to run relevant planned contrasts accordingly. By singularly comparing the aPFC group with all the other conditions (sham, OC, and dlPFC) we were able to observe that the CS-elicited fear was significantly reduced in the aPFC group (both in the test and the follow-up phases) relative to the conditioning phase. This dampening effect from the acquisition phase was selective for this region and thus was not observed in all the other groups.

– The generalization analyses should consist of the within subject factor Stimulus Type (CS, NS1, NS2) and between subject factor Group. Again, only in the case of a significant omnibus test, these should be followed up by relevant planned comparisons.

Thank you for this indication. We performed the suggested analysis by running 2 × 3 mixed ANOVAs with a within-subject factor of Stimulus Type/Tone (CS, NS_1_, and NS_2_) and a between-subject factor of Group (aPFC *vs* sham, aPFC *vs* OC, and aPFC *vs* dlPFC). This approach led to the confirmation of the previous findings that, during the test session, rTMS over the aPFC significantly decreased the defensive responses to all the stimuli (CS, NS_1_, and NS_2_) relative to the sham condition (we found a significant main effect of Group and no Group × Tone interaction effect), supporting a dampening effect of threat generalization processes. Finally, we confirmed the previous finding that the difference between the aPFC group and the dlPFC group pertaining to rTMS effects was selective for the CS, and no between-group differences were observed for defensive reactions to the NSs.

– Very important: to increase transparency and credibility, ideally, the paper should adopt a complete open science approach: the raw data along with the code to arrive at results should be openly accessible. The acquisition data should also be presented in graphs, both with all trials and the relevant averages that are chosen to compare for the pre-post intervention effect.

We agree with the Reviewer about the importance of transparency. To this aim, the raw data of this study have been uploaded along with the manuscript files. The methodological procedures to arrive at results have been detailed in the Methods section (“Psychophysiological recording and analysis” and “Statistical analyses” subsections).

– Generally, the factors and their levels of performed ANOVAs should always be reported.

As suggested, we always reported the factors and levels of performed ANOVAs in the Results section.

– Pay attention to using consistent numbers after the comma's, this currently varies widely.

Thank you for this advice. We have corrected the values throughout the manuscript, to uniform them with three numbers after the commas.

– For transparency, always report the exact p-value (so not p>0.05)

Thank you also for signaling this point. As recommended, we reported the exact p-value throughout the manuscript, also for the non-significant cases.

– To strengthen the theoretical framework of the paper, the rationale to include the explicit memory tests as well as a substantiation of the hypotheses should be included in the introduction. Also, a clear and substantial reasoning behind the inclusion of the generalization tests is required. It is now after the fact, but preregistration of the hypotheses and analytical approach would have been a major plus. Currently is seems a bit as if the study was actually set-up as some sort-of reconsolidation intervention. I am saying this because the introduction is oblivious as to per what mechanism could mediate the long-term effects of the intervention, if not extinction of reconsolidation. So a clear reasoning why the long-term test was implemented and why the authors hypothesize what they hypothesize should be added. For future studies, preregistration is highly recommended.

Thank you for these useful indications. As suggested, in the Introduction section we added the rationale to include the explicit memory test and a generalization test, the reasons to implement the long-term test, and the experimental hypotheses.

– The discussion (which is currently already interesting) could reflect more on what mechanisms may be behind the long-term effect, and how the absence of any effects of the explicit memory tests should be interpreted. A critical reflection on what exact theoretical constructs these tests represent is also required.

Also in this case, we added in the Discussion section a reasoning about the potential mechanisms mediating the long-term effect, the absence of any effects on the explicit memory, and the theoretical constructs that these tests represent (see also the Methods section).

Reviewer #2 (Recommendations for the authors):– Labeling all areas in the midline of the prefrontal cortex 'medial prefrontal cortex' might not be wrong, but lumping those very different regions together as a homogeneous functional entity can be a bit off-putting to those who consider anatomy important. Perhaps the authors can be more precise in their nomenclature throughout the manuscript.

The Reviewer is completely right about this point. To more precisely denominate the prefrontal target of our rTMS procedure, throughout the manuscript, we corrected “medial prefrontal cortex” (mPFC) with “medial anterior prefrontal cortex” (aPFC). Indeed, the coil placement that we adopted (i.e. over Fpz according to the international 10‒20 EEG coordinate system) directly focused on the medial portion of the BA 10.

– I would suggest to plot the SCR timeseries in addition to the deviation from baseline. That would also give insight into the amount of change caused by the TMS protocol.

Giving insights into the amount of change caused by the rTMS protocol is certainly of great importance. To this purpose, we have now added within-group comparisons (through 2 × 2 mixed ANOVAs) that show, for each group, the amount of change in CS-evoked SCRs from the conditioning phase to the test phase, as well as from the conditioning phase to the follow-up phase. Furthermore, to directly and simply depict these changes, in addition to dot plots, we have also represented them with line charts (Figures 2C, 2H, 4C, 4H, 5C, 5H).

– The sentence starting at lines 32-33: "in this scenario…" is quite difficult to understand, perhaps consider revising it.

Thank you. We have removed this unnecessary sentence.

– Lines 93-95 is quite confusing, for a moment I thought the authors were going to assess the potential distal effects of TMS, which did not happen. Consider keeping this for the discussion

Thank you. We have moved this part to the Discussion section.

– On several instances the authors equate a non-significant effect to effects being the same e.g. line 109-110: "there the CS evoked similarly strong autonomous reactions (P > 0.05) (Figure 2A)." That is not really what you can conclude from such an analysis as you do not test how similar these effects are. Additionally, it would be good to give the exact p-value and t-statistics for readers to judge how not different these effects are.

Thank you for this revision. We have corrected all the sentences like the one above-mentioned, and we have reported the exact p-value and statistics for non-significant comparisons throughout the manuscript.

– Consider adding simulations of the TMS effects on the neural tissue. That will make it easier for non-experts in TMS to judge the success of your manipulation.

Thank you for this interesting suggestion. We have performed simulations of the rTMS effects on the neural tissue of the medial anterior prefrontal cortex, the left occipital cortex, and the left dorsolateral prefrontal cortex. The simulations have been performed with SimNIBS 4.0 software, and they have been included in the main figures (Figure 2A, Figure 4A, and Figure 5A).

– It is not clear to me why the authors shift back to context A in their third session, is there a rationale for that?

The rationale behind the context shift between the second session (context B) and the third session (context A) consisted of testing potential renewal effects. Indeed, a *return of fear* following extinction training is often caused by a change of context, due to the fact that extinction learning is context-dependent (Vervliet et al., 2013). Therefore, we sought to test whether rTMS (delivered in context B) may induce enduring effects, observable even during a re-exposition to the original threatening environment (context A). To better clarify this point, we have included more explanations in the Introduction section, the Results section, and the Discussion section.

– Why is the time for the peak-to-trough deflection different for US2 as compared to the CS and US?

The reason why the time for peak-to-through deflection is different for the US_2_ compared to that of the other stimuli (CS, NSs) relies on the different duration of the selected stimuli (6s for the CS/NSs and 4s for the US_2_). Since we considered as event-related the SCRs in which the trough-to-peak deflection started at least 1s from the onset and before the offset of each stimulus, then the time window in which an event-related deflection had to begin was 1–6s (for the CS and the NSs) or 1–4s (for the US_2_) after the stimulus onset.

[Editors’ note: what follows is the authors’ response to the second round of review.]

The manuscript has been improved and the addition of control experiments has many of the concerns raised by reviewer 2 who has no further comments. However, there are some remaining issues related to the SCR data handling and reporting that need to be addressed. Reviewer 1 provides valuable concrete suggestions for analyses (analysing SCR data from all acquisition trials, bonferroni correction where appropriate) and reporting (discussing limitations related to efficacy of threat conditioning, the lack of a control stimulus CS-, failing generalisation manipulation that makes it unwarranted to claim that the intervention targeted generalization). An important possibility that needs to be discussed is that the effects reflect a general dampening effect on seeing any kind of stimulus (that is not aversive in itself).Reviewer #1 (Recommendations for the authors):The authors have revised the manuscript substantially, and the result certainly has improved. The revised version is clear, nuanced, and the contribution of the manuscript to the literature is evident. Some final considerations and concerns to reflect on to further improve the manuscript.

We would like to thank the Reviewer for the time and effort spent in carefully evaluating the revised version of the manuscript, and we are happy to read this recognition of our work. Thank you also for these useful considerations that you provided, which have been all addressed as follows:

- In the intro it would be good to add a clear and explicit rationale for the inclusion of the explicit memory test and the generalization test.

As suggested, we added in the Introduction (lines 38-41) a rationale for including the explicit memory test as well as the generalization test.

– Please also analyse the SCR data from ALL acquisition trials and add its graph on a trial-by-trial basis, this is the best way to assess any possible pre-existing learning differences for the groups.

Thank you for this important suggestion. As requested, we have analyzed the CS-related SCR data during the acquisition phase on a trial-by-trial basis ‒including both the 15 trials of the conditioning procedure separately as well as their average in the graphs (see Figure 2-S1, Figure 4-S1, and Figure 5-S1). Collectively, we found no differences in how the aPFC group responded relative to the sham, the OC, and the dlPFC groups, allowing us to exclude potential pre-existing learning differences between groups. These analyses have been included in the Results section, highlighted in yellow.

– One concern I have is the lack of a control stimulus (CS-) during acquisition: as such it is challenging to reveal whether conditioning was effective in the first place. In this light it is also concerning that the SCR values seem very low.

We thank the Reviewer for raising this important point. As we mentioned in the Results section (lines 101-102), we chose to adopt a single-cue conditioning protocol ‒instead of a differential conditioning procedure‒ because it more ecologically reflects real-life traumatic experiences. It is true that the consequent lack of a control stimulus (CS-) make it more challenging to reveal the efficacy and precision of the conditioning. However, one possibility to observe the efficacy of fear learning may consist of comparing the CS-evoked SCRs of the sham control group before the conditioning (preconditioning phase), during the conditioning, and after the conditioning (implicit test). In this way, it has been possible to observe a progressive enhancement of the magnitude of the autonomic responses through these three experimental phases. With the same logic, the aPFC group showed an enhancement of SCRs from the preconditioning to the conditioning phase, but a following decrease of SCRs after the rTMS procedure (during the implicit test) which restored the magnitude of CS-related responses to preconditioning levels.

– Please confirm that the shock electrodes were attached during the test phases and indicate so in the manuscript.

We confirm that the shock electrodes were attached during all the test and follow-up phases, and we have added this specification also for the US_2_ test and the perceptual test in the Methods section (lines 521-522 and 550), which were previously missing.

– It is unclear what tests were Bonferroni corrected and which ones were not (and why), this should be done consistently.

We confirm that we had performed Bonferroni-corrected tests throughout the manuscript, and now we have included this specification for all the *post-hoc* comparisons in the Results section. We apologize for the previous omission.

– The unit of SCR on some of the y-axes is lacking. On some other graphs it is the root of mS. The unit applied should be consistent across all graphs. In the graphs, also the relevant planned comparisons that are NS should be indicated.

We thank the Reviewer for bringing this issue to our attention. As requested, we have corrected the y-axes of the graphs and now the unit is consistent across all graphs. In the supplementary figures (Figure 2-S1, Figure 4-S1, and Figure 5-S1) we have also corrected the analysis of the CS-preconditioning (panel A) by processing the raw SCR data in the same way as all the other SCR data of the manuscript (in the previous version they were only square-rooted). The corrected analyses have been highlighted in yellow in the Results section. Furthermore, we have indicated also the relevant non-significant planned comparisons in the main figures.

– The lack of a main effect of tone in the generalization phase indicates that the generalisation manipulation itself did not work. This needs to be flagged in the relevant analyses and critically discussed in the Discussion section. For one thing, it does not allow the conclusion that the intervention targeted generalization. Consequently, one may wonder whether the effect (at least the immediate one, the follow-up test is another story) is mediated by memory at all, or whether it involves a more general dampening effect on seeing any kind of stimulus (that is not aversive in itself).

The Reviewer is absolutely right in raising this important point. Autonomic reactions that we observed towards the new tones in the aPFC group relative to the sham control group did not allow the conclusion that the rTMS intervention targeted threat generalization, leaving open the question of the specificity of rTMS effects (mediated by memory or more general dampening effect). However, the lack of between-group differences that we observed in the autonomic responses to the US_2_ seems to suggest that the observed effect may be memory-related and not due to a general dampening of autonomic reactivity. We added this reasoning in the Discussion section (lines 340-344).